# SWE-ABS: Adversarial Benchmark Strengthening Exposes Inflated Success Rates on Test-based Benchmark

**Boxi Yu** [* 1] **Yang Cao** [* 2] **Yuzhong Zhang** [3] **Liting Lin** [1] **Junjielong Xu** [3] **Zhiqing Zhong** [3] **Qinghua Xu** [1] **Guancheng Wang** [1] **Jialun Cao** [4 5] **Shing-Chi Cheung** [4 5] **Pinjia He** [† 3] **Lionel Briand** [1 6]

## Abstract

The SWE-Bench Verified leaderboard is approaching saturation, with the top system achieving 78.80%. However, we reveal that this performance is inflated: our re-evaluation demonstrates that one in five "solved" patches from the top-30 agents are semantically incorrect, passing only because weak test suites fail to expose their errors. We present SWE-ABS, an adversarial framework that strengthens test suites through a two-stage pipeline: (1) coverage-driven augmentation utilizing program slicing to target untested code regions, and (2) mutation-driven adversarial testing that synthesizes plausible-but-incorrect patches to expose semantic blind spots. On SWE-Bench Verified (500 instances), SWE-ABS strengthens 50.2% of instances (a $25.1\times$ improvement over prior work) and rejects 19.78% of previously passing patches. Consequently, the top agent's score decreases from 78.80% to 62.20%, causing significant leaderboard reshuffling (e.g., the top-ranked agent drops to 5th place).

## 1. Introduction

Reliable evaluation is fundamental to measuring progress in machine learning. As benchmarks become the primary mechanism for comparing system capabilities, ensuring they possess sufficient *discriminative power* (i.e., the ability to reject incorrect solutions while accepting correct ones) is critical. Yet we reveal a systematic evaluation crisis: test suites in widely used benchmarks, often derived from real-world repositories, carry over the limitations of their original test designs, allowing semantically incorrect solutions to pass and inflating success rates.

We demonstrate this crisis in SWE-Bench (Jimenez et al., 2024), a widely adopted benchmark for evaluating LLM-based software engineering agents. The top-ranked code agent achieves a 78.80% success rate, suggesting that the benchmark is approaching saturation. Such apparent saturation is not unique to SWE-Bench: recent evaluation research shows that binary test-based protocols can mask substantial quality gaps, as models may optimize for merely test-passing behavior under weak discriminative signals (Mang et al., 2025). However, our analysis shows that 19.78% of cases labeled as "solved" among the top-30 leaderboard agents are semantically incorrect. Across 11,041 patches produced by the top-30 leaderboard agents that pass the original SWE-Bench Verified tests, we reject 2,184 (19.78%) when applying strengthened test suites. This reduces the top agent's success rate by 16.6 percentage points, from 78.80% to 62.20%, causing it to drop from 1st to 5th place.

This crisis stems from a fundamental mismatch in objectives: SWE-Bench tests originate from development pull requests (PRs), which are designed to *verify if a specific patch passes the predefined test suites* rather than to *discriminate between all potential correct and incorrect solutions*. This creates two systematic weaknesses: *coverage gaps*, where tests miss patch-affected code entirely, and *semantic blind spots*, where tests accept superficially correct behavior without verifying deeper semantic requirements. Coverage gaps are straightforward to identify via static analysis; semantic blind spots, however, are more subtle and harder to detect.

To illustrate these weaknesses concretely, Figure 1 presents a real example from the SWE-Bench dataset where the original test suite fails to expose a semantic blind spot. The issue requires the PostgreSQL client backend to invoke `subprocess.run` for password retrieval, a function that requires a string argument. However, the original test only validates string passwords, creating a semantic blind spot: 25 patches that omit the `str()` conversion successfully pass the original tests but fail our augmented test, which exercises non-string inputs. As shown in Figure 1(b), a patch

---

* Equal contribution. † Corresponding author. [1]Lero the Research Ireland Centre for Software, University of Limerick [2]Independent Researcher [3]The Chinese University of Hong Kong, Shenzhen [4]Hong Kong University of Science and Technology [5]Guangzhou HKUST Fok Ying Tung Research Institute [6]University of Ottawa. Correspondence to: Pinjia He <hepinjia@cuhk.edu.cn>.

*Proceedings of the 43rd International Conference on Machine Learning*, Seoul, South Korea. PMLR 306, 2026. Copyright 2026 by the author(s).

### (a) Issue & Gold Patch

**Instance:** django__django-10973

**Issue:** Refactor the PostgreSQL client backend to use subprocess.run with the PGPASSWORD environment variable

**Gold Patch** ✓
```
subprocess_env = os.environ.copy()
if passwd:
  subprocess_env['PGPASSWORD']=str(passwd)
...
subprocess.run(args, env=subprocess_env)
```

*Correct:Explicit str() conversion ensures all environment variables are strings, satisfying subprocess.run's requirement.*

### (b) Original Test & Agent Patch

**Original Test Suite**
```
def test_basic():
  args, pgpassword =run(
      {'password':'secret'}
  )
  assert pgpassword == 'secret'
```

✗ **Semantic blind spot:**

Only string passwords are tested. Non-string inputs (e.g., integers) are never exercised.

**Agent Patch Example (TRAE + Doubao-Seed-Code)**
```
env = os.environ.copy()
if passwd:
  env['PGPASSWORD'] = passwd
```

**25 Wrong Patches PASS** ✓

### (c) Augmented Test

**After Test Enhancement**
```
def test_password_non_string():
  # Adversarial:  integer
  args, pgpassword =run(
      {'password':123456}
  )
  assert pgpassword =='123456'
```

**Impact:**

Exposes missing type conversion logic in environment variable handling.

**25 Wrong Patches FAIL** ✗

*Figure 1.* **Motivating example: Weak tests enable silent failures.** (a) A real Django issue (`django__django-10973`) refactoring requires converting passwords to strings before passing them to `subprocess.run` via the `PGPASSWORD` environment variable, because `subprocess.run` only accepts string-valued environment variables. (b) The original test suite tests only string passwords, enabling semantically incorrect agent patches to pass despite failing to handle non-string inputs. (c) A strengthened test introduces non-string inputs, exposing the missing type conversion and correctly failing the invalid patches.

generated by `TRAE + Doubao-Seed-Code` (Trae Research Team et al., 2025; ByteDance Seed et al., 2025), the current top-performing agent on SWE-Bench, is plausible but incorrect: it addresses the refactoring request yet misses the string-type constraint, thereby passing the original tests while violating the actual requirement.

**Adversarial Benchmark Strengthening.** Prior work, such as UTBoost (Yu et al., 2025), also attempts test augmentation by generating additional unit tests via LLMs, but strengthens only 10 out of 500 (2%) instances on SWE-Bench Verified. To uncover more vulnerabilities in the SWE-Bench test suite, we propose SWE-ABS, an *adversarial* framework that actively attacks test suites to expose latent weaknesses, and then fortifies them. Our two-stage approach combines complementary signals:

*Stage I: Coverage-Driven Augmentation* leverages program slicing to identify patch-affected code regions and generates tests to exercise them. A *test decoupling* mechanism is adopted to prevent generated tests from overfitting to the specific implementation details of the gold patch.

*Stage II: Mutation-Driven Adversarial Strengthening* targets subtle semantic blind spots. We synthesize "mutant" patches that are plausible (i.e., they pass all existing tests) yet semantically incorrect. By identifying mutants that evade existing tests, we generate targeted adversarial tests to reject them, effectively mirroring red-team/blue-team dynamics used in security testing.

**Results.** We evaluate SWE-ABS on SWE-Bench Verified (Chowdhury et al., 2024) (500 instances) and a subset of SWE-Bench Pro (Deng et al., 2025) (150 instances). On SWE-Bench Verified, SWE-ABS strengthens 50.2% of in-stances ($25.1\times$ improvement over UTBoost), and induces an average decline of 14.56 percentage points in resolve rates across systems, resulting in 30 rank changes within the top-30 leaderboard. Cross-benchmark evaluation reveals a notable observation: although current code agents achieve substantially lower resolve rates on SWE-Bench Pro than on SWE-Bench Verified (45.89% vs. 78.80% for the top system), SWE-ABS attains a comparable strengthening rate on both benchmarks. This indicates that benchmarks that are more challenging for models do not necessarily provide higher-quality or more discriminative test cases.

**Contributions.**

- **SWE-ABS**, a two-stage adversarial benchmark strengthening framework that combines coverage-driven augmentation with mutation-driven adversarial testing. SWE-ABS strengthens 50.2% of SWE-Bench Verified instances, a $25.1\times$ improvement over prior work (UTBoost).

- **Empirical evidence** that PR-driven test suites systematically lack discriminative power: our re-evaluation rejects 19.78% of previously accepted solutions (2,184 out of 11,041 patches) and induces 30 rank changes among the top-30 leaderboard agents.

- **A counterintuitive finding** that task difficulty and test discriminativeness are orthogonal: SWE-ABS achieves comparable strengthening rates on both SWE-Bench Verified and the more challenging SWE-Bench Pro.

- **Strengthened test suites** for SWE-Bench Verified (500 instances) and a subset of SWE-Bench Pro (150 instances), included in supplementary materials to facilitate more rigorous evaluation of future code agents.

## 2. Related Work

**Benchmarks for Code Generation.** Code generation evaluation has evolved from function-level benchmarks like HumanEval (Chen et al., 2021) and MBPP (Austin et al., 2021) to repository-level challenges. SWE-Bench (Jimenez et al., 2024) and its variants have progressively improved test validity (SWE-Bench Verified (Chowdhury et al., 2024)), language coverage (Multi-SWE-Bench (Zan et al., 2025)), contamination resistance (SWE-Bench Live (Zhang et al., 2025)), and task difficulty (SWE-Bench Pro (Deng et al., 2025)). SWE-Bench Pro curates more challenging instances from strong copyleft licenses codebases spanning Python, JavaScript, TypeScript, and Go, requiring larger code modifications (averaging 107 lines across 4 files). However, all SWE-Bench variants rely on PR-originated test suites designed to verify specific patches rather than discriminate among alternative solutions, leaving coverage gaps and semantic blind spots that allow incorrect patches to pass. Empirical studies further confirm that the existing test suites are too weak to reliably distinguish correct from incorrect patches (Aleithan et al., 2024). We select SWE-Bench Verified and SWE-Bench Pro as evaluation benchmarks to assess our method across both established and multilingual settings.

**Test Augmentation.** Test-based evaluation of program repair faces a fundamental challenge: patches that are plausible (passing all tests) may not be correct (genuinely fixing the bug), as they can overfit to insufficient test suites (Qi et al., 2015; Smith et al., 2015). EvalPlus (Liu et al., 2023) demonstrated test insufficiency in established benchmarks, augmenting HumanEval (Chen et al., 2021) and MBPP (Austin et al., 2021) with $80\times$ more tests. UT-Boost (Yu et al., 2025) pioneered test augmentation for SWE-Bench, exposing 345 previously undetected incorrect patches on SWE-Bench. Recent work further explores LLM-based test generation via multi-agent collaboration (Xu et al., 2026; Wang et al., 2026c) or coverage-guided prompting (**?**Ryan et al., 2024). PatchDiff (Wang et al., 2025) generates differential tests by contrasting an agent patch against the gold patch, but requires re-running per agent submission. However, these approaches generate tests without explicitly analyzing patch-relevant code regions or targeting semantic correctness, limiting their ability to systematically expose both coverage gaps and semantic blind spots.

**Mutation Testing.** Mutation testing (DeMillo et al., 1978; Jia & Harman, 2011) evaluates test quality by introducing artificial faults, traditionally using syntactic operators (e.g., replacing + with −). Recent work leverages LLMs to generate more realistic mutants for the evaluation of test adequacy (Wang et al., 2026a) and the generation of targeted tests (Dakhel et al., 2023; Harman et al., 2025; Wang et al., 2026b). SWE-ABS extends this paradigm by synthesizing LLM-based semantic mutations that represent plausible but incorrect patches that pass existing test suites but violate actual requirements, then generating tests that expose these blind spots.

## 3. Method

SWE-Bench relies on developer-written tests that provide necessary but insufficient conditions for patch correctness: a correct patch must pass these tests, but passing does not guarantee the correctness. As discussed in Section 1, this leads to two systematic weaknesses: *coverage gaps*, where tests fail to exercise patch-affected code regions, and *semantic blind spots*, where tests accept superficially correct behavior without verifying deeper semantic requirements.

We address these complementary weaknesses through a two-stage adversarial strengthening framework (Figure 2). Stage I targets coverage gaps by using program slicing to identify patch-relevant code and generating tests that exercise those regions, while decoupling these tests from gold-patch-specific behaviors to avoid overfitting. Stage II targets semantic blind spots by synthesizing plausible but incorrect mutant patches that evade existing tests, then generating adversarial tests to reject them.

Each SWE-Bench instance consists of an issue description $I$, a gold patch $P_{\text{gold}}$, and an original test suite $T_{\text{ori}} = T_{\text{base}} \cup T_{\text{patch}}$, where $T_{\text{base}}$ denotes the repository's existing tests and $T_{\text{patch}}$ denotes the tests added in the pull request. Our goal is to construct an augmented test suite $T_{\text{aug}}$ that rejects incorrect patches while accepting correct alternative implementations.

### 3.1. Stage I: Coverage-Driven Test Augmentation

Coverage gaps arise when tests fail to exercise patch-affected code regions, allowing incorrect patches to pass undetected. This stage addresses such gaps through four steps: (1) initial test generation creates tests targeting the issue, (2) test decoupling removes gold-patch-specific implementation details, (3) program slicing identifies patch-relevant code regions $L_{\text{rel}}$, and (4) coverage-guided enhancement adds tests for uncovered lines in $L_{\text{rel}}$.

#### 3.1.1. GENERATING INITIAL TESTS

In the first step, we prompt the LLM with the issue $I$, gold patch $P_{\text{gold}}$, and patch tests $T_{\text{patch}}$ to synthesize the initial augmented test suite $T_{\text{initial}}$ (all prompt templates can be found in Appendix A.1). The model is instructed to generate diverse test cases that cover corner cases. Providing $T_{\text{patch}}$ as context helps it understand the intended test style.

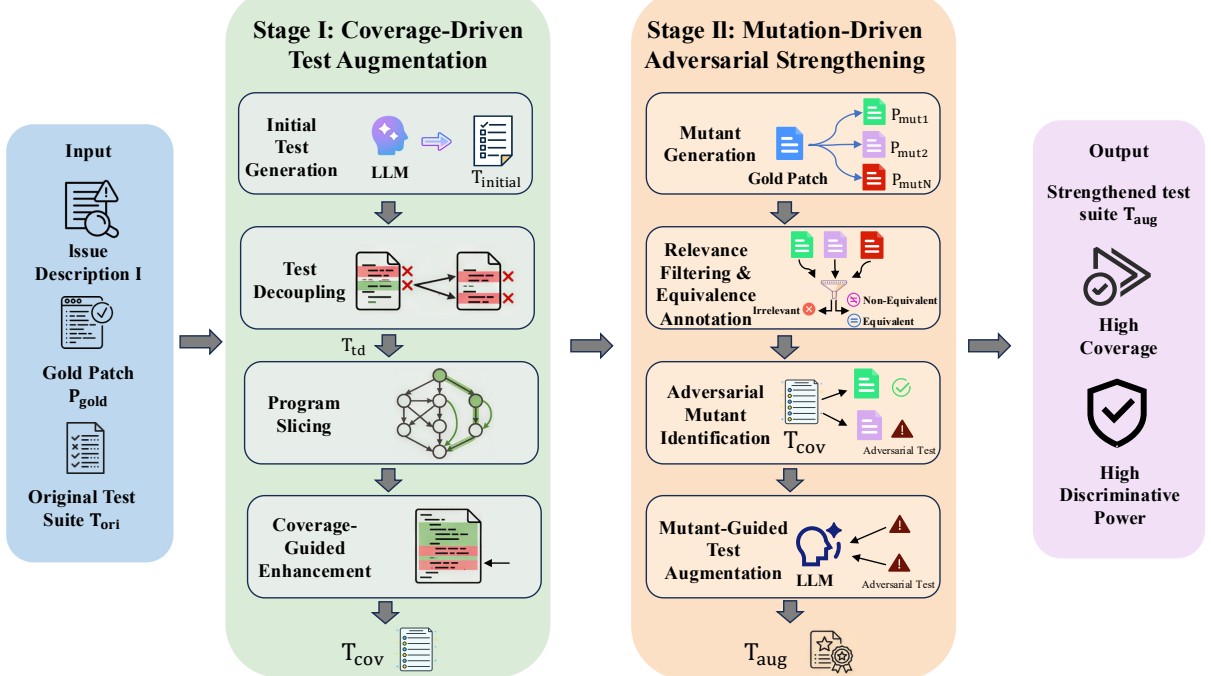

*Figure 2.* Overview of the SWE-ABS framework.

### 3.1.2. DECOUPLING GOLD-PATCH DEPENDENCIES

While initial test generation produces diverse tests, some may be overly specific to the gold patch, binding to its implementation-specific details and rejecting semantically correct alternative patches rather than enforcing the behavior required to resolve the issue (formal definition and an example in Appendix A.2). To mitigate this issue in $T_{\text{initial}}$, we introduce a test decoupling module. We first prompt the LLM to detect the tests that are over-specialized to gold patch, such as hard-coded error messages or implementation-specific side effects, which may reject some valid alternative fixes. These problematic tests are then refined by prompting the LLM to generalize them, ensuring that their core purpose is to verify whether the original issue has been resolved, rather than enforcing a particular implementation. After test decoupling, we verify whether the gold patch still passes the refined tests. We define the subset of refined tests that the gold patch passes as $T_{\text{td}}$.

### 3.1.3. PROGRAM SLICING

To target test generation toward patch-relevant code, we use intraprocedural program slicing (details in Appendix A.4.2). We construct a program dependence graph that captures data and control dependencies, then compute patch-relevant lines $L_{\text{rel}}$ as the forward and backward slice from modified lines, capturing both upstream dependencies and downstream dependents. To balance precision and scalability, we restrict analysis to statements within the same function or class as the patch, avoiding prohibitively expensive whole-program

analysis.

### 3.1.4. COVERAGE-GUIDED AUGMENTATION

With patch-relevant lines identified, we now enhance coverage of these regions. For each test $t \in T_{\text{td}}$, we measure its coverage over the patch-relevant lines $L_{\text{rel}}$ by using a code coverage tool, where the details are in Appendix A.4.3. We feed the identified uncovered lines to the LLM and prompt it to generate tests that address the uncovered code branches. We define the coverage-guided enhanced test as follows:

$$T_{\text{cov}} = T_{\text{td}} \ \cup \ \text{LLM}_{\text{cov}}\Big( L_{\text{rel}} \setminus \bigcup_{t \in T_{\text{td}}} \text{Cov}(t) \Big),$$

where $\text{Cov}(t) \subseteq L_{\text{rel}}$ denotes the set of lines exercised by test $t$, and $\text{LLM}_{\text{cov}}$ generates tests to cover the specified lines. This coverage-guided enhancement aims to maximize coverage of patch-relevant lines.

### 3.2. Stage II: Mutation-Driven Adversarial Strengthening

Stage I produces $T_{\text{cov}}$, which ensures *reachability* of patch-relevant code. However, reachability alone does not guarantee *observability* of semantic faults: a faulty implementation may execute the same control-flow paths as the correct patch yet produce incorrect states or outputs that existing assertions fail to detect. To address this limitation, Stage II employs mutation-driven adversarial strengthening to target *semantic blind spots*, i.e., scenarios where incorrect implementations satisfy coverage requirements but fail to

correctly resolve the issue.

### 3.2.1. MUTANT GENERATION

We employ an LLM-based mutation generator, denoted as $\text{LLM}_{\text{mut}}$, to produce mutant patches $P_{\text{mut}}$. Given a gold patch $P_{\text{gold}}$, $\text{LLM}_{\text{mut}}$ generates mutants that introduce subtle semantic faults while passing the original test suite $T_{\text{ori}}$:

$$P_{\text{mut}} = \text{LLM}_{\text{mut}}(P_{\text{gold}}).$$

### 3.2.2. RELEVANCE FILTERING AND EQUIVALENCE ANNOTATION

To mitigate hallucinated or irrelevant mutations, we apply an LLM-based filtering module that evaluates each mutant along two orthogonal dimensions: (1) *issue relevance*, i.e., whether the mutation targets behaviors or code regions implicated by the original issue, and (2) *semantic equivalence*, i.e., whether the mutation is functionally equivalent to the gold patch. For each dimension, we obtain independent judgments from $k$ LLMs (with $k = 3$ in our experiments) and apply majority voting to determine the final label. A comparison with human annotations, along with illustrative examples, is provided in the Appendix A.5. We first filter out issue-irrelevant mutants to obtain a subset of issue-relevant mutants $P_{\text{rel}} \subseteq P_{\text{mut}}$. Then, we label the semantic equivalence of each patch $p$, where $\text{equiv}(p)$ denotes semantic equivalence and $\neg\text{equiv}(p)$ denotes semantic non-equivalence.

### 3.2.3. IDENTIFYING ADVERSARIAL MUTANTS

After obtaining the filtered and annotated mutants $P_{\text{rel}}$, we identify which mutants expose weaknesses in the current test suite $T_{\text{cov}}$ by checking whether each mutant $p$ passes or fails the tests. An ideal test suite should accept semantically equivalent mutants and reject non-equivalent ones. Discrepancies between expected and actual outcomes reveal test suite weaknesses:

- **False negatives**: Equivalent mutants (correct alternatives) that fail $T_{\text{cov}}$, indicating tests are overly specific to the gold patch.

- **False positives**: Non-equivalent mutants (incorrect patches) that pass $T_{\text{cov}}$, revealing semantic blind spots.

We partition these discrepancy-inducing mutants into two categories based on their failure mode:

$$P_{\text{adv}}^{\text{FN}} = \{p \in P_{\text{rel}} \mid \text{equiv}(p) \wedge p \not\models T_{\text{cov}}\}, \quad (1)$$
$$P_{\text{adv}}^{\text{FP}} = \{p \in P_{\text{rel}} \mid \neg\text{equiv}(p) \wedge p \models T_{\text{cov}}\}. \quad (2)$$

For each false-negative mutant $p \in P_{\text{adv}}^{\text{FN}}$, we further identify the specific tests responsible for the incorrect rejection:

$$T_{\text{fn}} = \bigcup_{p \in P_{\text{adv}}^{\text{FN}}} \{t \in T_{\text{cov}} \mid p \not\models t\}.$$

### 3.2.4. MUTANT-GUIDED TEST AUGMENTATION

**Fixing False Negatives.** For tests in $T_{\text{fn}}$ that incorrectly reject equivalent mutants, we invoke an LLM-based test fixer $\text{LLM}_{\text{fix}}$ to generalize them:

$$T_{\text{fn}}' = \text{LLM}_{\text{fix}}\left(T_{\text{fn}}, P_{\text{adv}}^{\text{FN}}\right).$$

The fixer relaxes overly specific assertions while preserving the tests' ability to verify the original issue resolution.

**Fixing False Positives.** For each false-positive mutant $p \in P_{\text{adv}}^{\text{FP}}$, we invoke an LLM-based test generator $\text{LLM}_{\text{aug}}$ to produce targeted tests that reject $p$:

$$T_{\text{mut}}(p) = \text{LLM}_{\text{aug}}(p, T_{\text{cov}}).$$

**Final Test Suite.** We construct the final augmented test suite by replacing the overly strict tests with their generalized versions and adding the new discriminative tests:

$$T_{\text{aug}} = (T_{\text{cov}} \setminus T_{\text{fn}}) \cup T_{\text{fn}}' \cup \bigcup_{p \in P_{\text{adv}}^{\text{FP}}} T_{\text{mut}}(p).$$

This adversarial augmentation complements the coverage-driven tests from Stage I, yielding a test suite that is both comprehensive in coverage and robust against plausible but incorrect fixes.

## 4. Experiments

Our experiments aim to answer the following research questions:

- **RQ1 (Effectiveness):** Does SWE-ABS significantly improve the discriminative power of SWE-Bench test suites in distinguishing correct fixes from incorrect patches?

- **RQ2 (Generalization):** Does SWE-ABS generalize across different benchmarks and underlying base models?

- **RQ3 (Test Quality):** Do SWE-ABS-augmented tests maintain robustness to alternative correct implementations while effectively detecting semantic errors?

- **RQ4 (Ablation Study):** How do the coverage-driven and mutation-driven stages individually and jointly contribute to effectiveness?

*Table 1.* Agent-level results on SWE-Bench Verified (Top-5 systems) for TRAE (Trae Research Team et al., 2025) with Doubao-Seed-Code (ByteDance Seed et al., 2025), Live-SWE-agent (Xia et al., 2025) with Gemini (Gemini Team, 2025) 3 Pro Preview, 2025-11-18, Atlassian Rovo Dev (Atlassian, 2025) (2025-09-02), EPAM AI/Run Developer Agent (EPAM Systems, Inc., 2025) (v20250719) with Claude 4 Sonnet and ACoder (ACoder Team et al., 2025). Scores denote resolve rates (%) under the *original* (SWE-Bench Verified), UTBoost, and SWE-ABS test suites. Drop denotes the per-agent decrease in resolve rate (in percentage points). Rank indicates changes in leaderboard position among all 30 agents. Full results are in Appendix A.6.

| Agent | Original | UTBoost | | | SWE-ABS (Ours) | | |
|---|---|---|---|---|---|---|---|
| | Score | Score | Drop | Rank | Score | Drop | Rank |
| TRAE + Doubao-Seed-Code | 78.80 | 77.80 | 1.00 | 1→1 | 62.20 | 16.60 | 1→5 |
| live-SWE-agent + Gemini 3 Pro Preview (2025-11-18) | 77.40 | 76.80 | 0.60 | 2→2 | 65.40 | 12.00 | 2→1 |
| Atlassian Rovo Dev (2025-09-02) | 76.80 | 75.60 | 1.20 | 3→5 | 61.20 | 15.60 | 3→8 |
| EPAM AI/Run Developer Agent v20250719 + Claude 4 Sonnet | 76.80 | 76.20 | 0.60 | 4→3 | 63.80 | 13.00 | 4→3 |
| ACoder | 76.40 | 75.80 | 0.60 | 5→4 | 61.80 | 14.60 | 5→6 |

*Table 2.* Benchmark-level results on SWE-Bench Verified. Str. (Strengthened) denotes the number of instances where at least one previously passing patch fails under augmented tests. Avg. Drop is measured in percentage points. Patch Kill denotes the total number of patches rejected by augmented tests. # Rank Changes denotes the number of agents whose leaderboard position changed. Spearman $\rho$ indicates the rank correlation coefficient between old and new leaderboard positions ($\rho = 1$ indicates perfect stability). SWE-ABS* excludes the 53 instances with overfitting tests (Section 4.4.1); SWE-ABS reports results after fixing the overfitting tests in those instances.

| Augmentation | Str. | Avg. Drop | Patch Kill | # Rank Changes | Spearman $\rho$ |
|---|---|---|---|---|---|
| UTBoost | 10 / 500 | 0.70 | 105/11,041 | 24 | 0.98 |
| SWE-ABS* | 206 / 500 | 11.22 | 1683/11,041 | 25 | 0.86 |
| SWE-ABS | 251 / 500 | 14.56 | 2184/11,041 | 30 | 0.82 |

## 4.1. Experimental Setup

**Benchmarks and Evaluation Protocol.** We evaluate on two benchmarks: **SWE-Bench Verified** (Chowdhury et al., 2024), containing 500 human-validated Python instances where the current best agent achieves 78.80% resolve rate; and **SWE-Bench Pro** (Deng et al., 2025), a harder, contamination-resistant benchmark with 731 multi-language instances (Python, JavaScript, Go, TypeScript) from strong copyleft licenses repositories, where the best code agent achieves only 45.89%. From SWE-Bench Pro, we stratify by programming language and randomly sample 150 instances from those for which at least one leaderboard agent produces a patch that passes the original tests. A per-language breakdown of the SWE-Bench Pro sampling distribution is provided in Table 12 (Appendix A.9). For each instance, we collect patches from leaderboard agents that pass the original tests (top-30 for SWE-Bench Verified, all 13 available for SWE-Bench Pro) and re-evaluate them under augmented tests. Leaderboard submissions and patch data were collected as of December 31, 2025.

**Evaluation Metrics.** We report four primary metrics: (1) **Strengthened (Str.)**: the number of instances where at least one previously passing patch fails under augmented tests; (2) **Drop**: the decrease in resolve rate (in percentage points); (3) **Patch Kill**: the total number of patches rejected by augmented tests; (4) **Spearman** $\rho$: the rank correlation coefficient between the original and augmented leaderboard

rankings, measuring the extent of ranking stability ($\rho = 1$ indicates perfect stability, while lower values indicate greater reordering; detailed calculation in Appendix A.4.4).

**Model Usage.** We use GPT-5 (OpenAI, 2025) as the default base model throughout all stages of SWE-ABS. To demonstrate generalizability, we also conduct controlled experiments with an alternative open-source model, GLM-4.7 (GLM Team, 2025). Unless otherwise specified, all reported results are obtained using GPT-5.

**Patch-Relevant Code Identification.** We perform intraprocedural program slicing using Tree-sitter (Tree-sitter, 2026) to identify code regions that are data-flow or control-flow dependent on a given patch (details in Appendix A.4.2).

All hyperparameters used in SWE-ABS, including temperature and maximum number of tries, are provided in Appendix A.4. All key prompts used for SWE-ABS are provided in Appendix A.1.

## 4.2. RQ1: Effectiveness on SWE-Bench Verified

We compare SWE-ABS against UTBoost (Yu et al., 2025) on all 500 SWE-Bench Verified instances. Since none of the top-30 agents used in our evaluation were included in UTBoost's original study, the UTBoost numbers reported here are obtained by re-running its publicly released augmented test suite on this new agent pool, and therefore differ from the figures cited in the original UTBoost paper.

**Results.** As shown in Tables 1 and 2, SWE-ABS substantially outperforms UTBoost across all metrics. At the benchmark level, SWE-ABS strengthens 251/500 instances (50.2%) compared to only 10 (2%) under UTBoost. Moreover, it identifies and rejects 2,184 of the 11,041 previously passing patches (19.78%) as insufficiently correct. Detailed per-repository breakdowns are provided in Table 11 (Appendix A.9). Table 2 also includes SWE-ABS*, which simply excludes the 53 overfitting instances identified in Section 4.4.1 without correction; SWE-ABS reports the final results after those overfitting tests are fixed.

At the agent level, SWE-ABS induces an average resolve-rate drop of 14.56 percentage points, compared to just 0.70 percentage points for UTBoost. This drop triggers significant leaderboard reordering, exposing evaluation instability under weak test suites. The Spearman rank correlation $\rho$ decreases from 0.98 (UTBoost) to 0.82 (SWE-ABS), indicating substantial ranking instability when test suites are properly strengthened. Among the top-5 agents, four experience significant rank changes under SWE-ABS (Table 1). Notably, TRAE (Trae Research Team et al., 2025) drops from 1st to 5th despite achieving the highest original score (78.80%), suggesting that its patches, while strong in many cases, may fail to handle certain corner cases effectively. In contrast, live-SWE-Agent (Xia et al., 2025) climbs from 2nd to 1st, indicating more robust and reliable patch generation overall. This instability underscores that current leaderboard rankings may not reflect true agent capabilities.

The full Top-30 leaderboard changes, along with results for Bash-only agents on SWE-Bench Verified, are reported in Appendix A.6 and Appendix A.7, respectively. A detailed cost analysis in Appendix A.10 shows that SWE-ABS achieves $16\times$ better cost efficiency than UTBoost ($4.98 vs. $80.00 per strengthened instance).

An additional comparison against the concurrent PatchDiff approach (Wang et al., 2025), which requires per-agent test regeneration, is provided in Appendix A.8; SWE-ABS attains a comparable resolve-rate drop with a single reusable test suite.

### 4.3. RQ2: Generalization

#### 4.3.1. CROSS-BENCHMARK GENERALIZATION

To verify that SWE-ABS generalizes beyond Python-only SWE-Bench Verified, we evaluate on SWE-Bench Pro, a harder, contamination-resistant benchmark with multi-language instances (Python, JavaScript, Go, TypeScript).

**Results.** Table 3 shows that SWE-ABS consistently strengthens test suites across benchmarks with substantially different difficulty levels. Despite the top-system resolve rate on SWE-Bench Pro being 33 points lower than on SWE-

Bench Verified (45.89% vs. 78.80%), SWE-ABS induces comparable strengthening effects: the average resolve-rate drop is 16.46 percentage points on the SWE-Bench Pro subset vs. 14.56 percentage points on SWE-Bench Verified, indicating that the discriminative power of augmented tests generalizes.

*Table 3.* Cross-benchmark validation. Statistics are computed on our sampled subset.

| Benchmark | Str. | Avg. Drop |
|---|---|---|
| SWE-Bench Verified | 251 / 500 (50.2%) | 14.56 |
| SWE-Bench Pro | 97 / 150 (64.7%) | 16.46 |

This validates two key insights: (1) **Task difficulty $\neq$ test strength**: a benchmark can be hard yet still have weak test suites; (2) **Method generalization**: SWE-ABS addresses fundamental test inadequacy that persists across benchmarks, not artifacts specific to Verified. The consistent effectiveness on contamination-resistant SWE-Bench Pro instances further mitigates data leakage concerns. We further report per-language breakdowns on SWE-Bench Pro in Table 12 (Appendix A.9)

#### 4.3.2. BASE MODEL GENERALIZATION

To evaluate the generalizability of SWE-ABS beyond GPT-5, we conduct a controlled experiment with the open-source model GLM-4.7, keeping all non-model components identical. SWE-ABS is applied to a random subset of 150 instances from SWE-Bench Verified under both model configurations.

**Results.** As shown in Table 4, SWE-ABS exhibits comparable strengthening behavior across the two base models. Using GLM-4.7 yields a comparable number of strengthened instances (81 vs. 81 out of 150) and a slightly lower average drop in resolve rate (16.36 percentage points vs. 16.60 percentage points) compared to GPT-5. These results suggest that the effectiveness of SWE-ABS is not tightly coupled to a specific proprietary base model and that the framework maintains its discriminative impact when instantiated with an alternative open-source model.

*Table 4.* Base model generalization. Evaluated on 150 randomly sampled SWE-Bench Verified instances.

| Base Model | Str. | Avg. Drop |
|---|---|---|
| GPT-5 | 81 / 150 (54%) | 16.60 |
| GLM-4.7 | 81 / 150 (54%) | 16.36 |

### 4.4. RQ3: Test Quality

We analyze the quality of SWE-ABS-augmented tests along two complementary dimensions: **(1) robustness** to alter-

native correct implementations (avoiding false rejections), and **(2) semantic error coverage** (characterizing true rejections).

#### 4.4.1. ROBUSTNESS TO ALTERNATIVE CORRECT FIXES

A key concern is whether aggressive test-suite strengthening leads to overfitting to the gold patch, thereby inadvertently rejecting valid alternative fixes.

Manual inspection of all 500 SWE-Bench Verified instances revealed that 53 contain overfitting tests encoding gold-patch-specific behavior (see an example in Appendix A.11.1), while the remaining 447 reflect legitimate strengthening that correctly filter out spurious fixes. For these 53 overfitting cases, we correct the augmented tests to revise or remove the gold-patch-specific assertions; Tables 1 and 2 report results after this correction. Overall, SWE-ABS achieves a conservative false-negative rate of 10.6% (53/500).

#### 4.4.2. SEMANTIC ERROR COVERAGE

To understand what errors SWE-ABS detects, we manually analyzed 100 agent-generated patches rejected by $T_{\text{aug}}$ but accepted by $T_{\text{ori}}$. Two authors independently categorized root causes, resolving disagreements through discussion; the Cohen's $\kappa$ on failure-mode classification is 0.67 (substantial agreement), with disagreements occurring mainly at the boundary between logic errors and incomplete fixes.

Table 5 summarizes the distribution of error types; representative examples illustrating each category are provided in Appendix A.11.2.

*Table 5.* Error taxonomy. Distribution of error types in 100 sampled patches rejected by SWE-ABS but accepted by original tests.

| Error Category | Count (%) |
| --- | --- |
| Logic errors | 46 (46%) |
| Incomplete fixes | 29 (29%) |
| Type mismatches | 13 (13%) |
| Boundary violations | 12 (12%) |

**Logic errors** (46%) dominate, producing patches that pass the original test suite yet implement fundamentally incorrect logic, leading to failures on unseen scenarios. **Incomplete fixes** (29%) are also frequent: patches resolve the primary case while overlooking related call sites or branches, suggesting that AI systems tend to generate "shallow" solutions that satisfy explicit tests without achieving semantic completeness. Type mismatches (13%) and boundary violations (12%) account for the remaining 25% of failures.

**Implications.** The dominance of logic errors (46%) and incomplete fixes (29%) reveals a systematic weakness in current AI code generation: agents tend to produce "shallow"

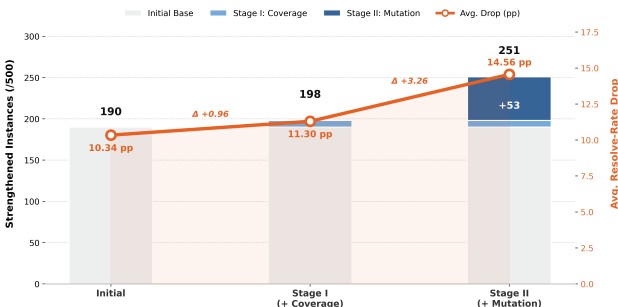

*Figure 3.* Ablation study on the effect of coverage-driven and mutation-based augmentations. Initial denotes baseline tests generated without any augmentation stage.

solutions that satisfy explicit test cases without achieving semantic completeness. This behavior aligns with training paradigms that optimize for test-passing reward signals, inadvertently encouraging agents to "teach to the test" rather than reason about program semantics.

The low prevalence of syntactic errors (type mismatches and boundary violations together account for 25%) suggests that modern LLMs have largely solved surface-level code correctness. The remaining challenge lies in *semantic reasoning*, which means understanding what the code should do, not just what makes tests pass. This finding has implications for future benchmark design: effective evaluation requires tests that probe semantic understanding rather than just syntactic correctness.

### 4.5. RQ4: Ablation Study

**Coverage-Driven Augmentation (Stage I).** We first evaluate the effect of applying only the coverage-driven augmentation stage. This stage aims to improve code reachability by identifying patch-relevant program regions via slicing and generating additional tests targeting previously uncovered code paths. As shown in Figure 3, this strategy yields a modest improvement on both reported metrics compared to the initial generated tests. The number of strengthened instances increases from 190 to 198, while the average drop in resolve rate improves from 10.34 to 11.30 percentage points. These results indicate that exercising patch-relevant but previously uncovered code regions helps reveal certain semantic faults that were masked due to insufficient execution. However, the limited gains suggest that reachability alone is insufficient to detect subtle bugs when existing assertions do not adequately constrain program behavior.

**Mutation-Driven Adversarial Strengthening (Stage II).** We further assess the contribution of the mutation-driven adversarial strengthening stage by incorporating it on top of coverage-driven augmentation. This stage explicitly targets semantic blind spots by synthesizing plausible but incorrect

mutant patches and generating tests designed to distinguish them from correct patches. With the second stage, SWE-ABS achieves substantial additional gains. In particular, the number of strengthened instances increases to 251, representing 44 more instances than Stage I alone. Meanwhile, the average drop in resolve rate rises significantly to 14.56 percentage points, corresponding to an absolute improvement of 3.21 points over Stage I. These results demonstrate that mutation-driven adversarial testing effectively complements coverage enhancement by uncovering semantic weaknesses that persist even when patch-relevant code is fully exercised.

**Complementarity Analysis.** Taken together, the ablation results suggest that coverage- and mutation-driven augmentations operate along *complementary dimensions* of test adequacy. Coverage augmentation primarily enforces execution over structurally relevant code regions, improving structural adequacy, while mutation-driven augmentation exposes semantically incorrect behaviors that remain executable yet violate intended program semantics.

**Additional Ablation: Test Decoupling.** The two-stage ablation above holds the upstream test-decoupling submodule (Section 3.1.2) fixed. A separate, finer-grained ablation in Appendix A.3 shows that disabling decoupling on a 100-instance subset nearly doubles the number of overfit cases ($9 \rightarrow 17$), confirming that the additional rejections obtained without decoupling largely reflect overfitted tests rather than genuine strengthening.

## 5. Conclusions, Limitations, and Future Work

We present SWE-ABS, a framework for adversarial benchmark strengthening that systematically augments test suites through coverage-driven generation and mutation-driven strengthening. Applied to SWE-Bench Verified, our approach reveals that 19.78% of accepted patches are semantically incorrect, strengthens 50.2% of instances ($25.1\times$ over prior work), and reduces the top agent's score from 78.80% to 62.20%. Cross-benchmark analysis further demonstrates that task difficulty and test strength are orthogonal, suggesting current leaderboard rankings may not reflect true code agent capabilities.

Our approach has several limitations: (1) *Test overfitting risk*: augmented tests may encode gold-patch-specific behaviors, causing a 10.6% false-negative rate; (2) *Analysis scope*: intraprocedural slicing may miss cross-module dependencies; (3) *Gold patch dependency*: requiring reference implementations limits applicability to scenarios without them. We discuss these limitations in detail in Appendix A.12.

Looking ahead, we identify several promising directions: (1) extending language coverage to Java, C++, and

Rust through language-specific AST parsers; (2) integrating SWE-ABS into continuous benchmark co-evolution pipelines that periodically re-strengthen test suites; and (3) leveraging strengthened test suites as training signals in environments such as SWE-Gym (Pan et al., 2025).

We believe SWE-ABS establishes a principled methodology for adversarial benchmark strengthening in code generation and repair evaluation, with potential applicability to other domains with executable oracles.

## Software and Data

All data, enhanced test suites, and evaluation scripts are publicly available at: https://github.com/OpenAgentEval/SWE-ABS.

## Acknowledgements

This publication has emanated from research conducted with the financial support of Taighde Éireann – Research Ireland under Grant Number. 13/RC/2094_2. This work was also supported by the Guangdong Basic and Applied Basic Research Foundation (Grant No. 2024A1515010145), the National Natural Science Foundation of China (Grant No. 92582201), and the Research Grants Council through the General Research Fund (Ref No. 16210725) and the Theme-based Research Scheme (Ref No. T41-517/25-N).

## Impact Statement

This work addresses a fundamental challenge in evaluating AI code generation systems: ensuring that benchmarks provide reliable signals of system capabilities rather than inflated success rates that obscure true performance.

**Reliable Evaluation Enables Informed Decisions.** By demonstrating that 19.78% of patches previously labeled as "solved" are semantically incorrect, we reveal that weak test suites can mislead both researchers prioritizing future directions and practitioners making deployment decisions. Our cross-benchmark analysis further shows that task difficulty and test strength are orthogonal: harder benchmarks do not automatically provide more discriminative evaluations. This underscores the need for principled test strengthening as a distinct concern.

**Implications for AI Safety.** As AI coding assistants are increasingly deployed in production environments, rigorous evaluation becomes a safety concern. Patches that pass weak tests may omit critical input validation or error handling, potentially introducing vulnerabilities when deployed. Strengthened benchmarks help surface such failure modes before real-world deployment.

**Broader Applicability.** While demonstrated on program

repair, the adversarial strengthening methodology applies to any domain where correctness can be verified through automated testing, including code translation, text-to-SQL, and robotic control. As AI systems are increasingly evaluated via test-based benchmarks, ensuring sufficient discriminative power becomes essential for meaningful progress measurement.

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

# A. Appendix

The appendix is organized as follows:

- **Appendix A.1** lists all prompt templates used in the pipeline.

- **Appendix A.2** provides the formal definition of overfit tests and accompanying case studies.

- **Appendix A.3** ablates the test-decoupling module.

- **Appendix A.4** details experimental implementation including hyperparameters and program slicing.

- **Appendix A.5** covers mutation cases and evaluation.

- **Appendix A.6** provides full leaderboard results for top-30 agents on SWE-Bench Verified.

- **Appendix A.7** provides full leaderboard results for Bash-only agents on SWE-Bench Verified.

- **Appendix A.8** provides a head-to-head comparison with the concurrent PatchDiff approach.

- **Appendix A.9** reports additional experimental results by repository and language.

- **Appendix A.10** presents computational cost analysis.

- **Appendix A.11** presents qualitative case studies of test strengthening failures and agent-generated patch failures.

- **Appendix A.12** discusses limitations and threats to validity.

## A.1. Prompt Templates

### A.1.1. STAGE I: INITIAL TEST GENERATION PROMPT

```
You are a helpful assistant that can interact multiple times with a computer shell to
    solve programming tasks.
Your response must contain exactly ONE bash code block with ONE command (or commands
    connected with && or ||).

Include a THOUGHT section before your command where you explain your reasoning process.
Format your response as shown in <format_example>.

<format_example>
THOUGHT: Your reasoning and analysis here

```bash
your_command_here
```
</format_example>

Failure to follow these rules will cause your response to be rejected.

<pr_description>
Consider **the** following PR description:
{{task}}
</pr_description>

<instructions>
# Task Instructions

## Overview
You're a software engineer interacting continuously with a computer by submitting
    commands.
Your task is to reproduces **the** issue by writing independent test case and cover as
    many corner cases as possible.
```

```
There is a gold patch that is used to fix the issue.
--- Gold Patch ---
```
{{gold_patch}}
```

There is an original test patch that is used to verify the fix. You can learn from it
    to generate new test cases that thoroughly test the fix.
--- Original Test Patch ---
```
{{test_patch}}
```

IMPORTANT:
- This is an interactive process where you will think and issue ONE command, see its
    result, then think and issue your next command.
- {{workdir}} is the working directory for all your subsequent commands.
- Gold patch is already apply to the given repository.
- Test execution command example in this repository: {{test_command}}
- Write the new test case in a new test file, do not modify the original test file!!!
- Do not use other test command, do not run the full test suite

## Recommended Workflow
1. Understand the Problem:
   (1) Begin by carefully reading the user's problem description to fully grasp the
     issue.
   (2) Identify the core components and expected behavior.
2. Explore and Locate:
   (1) Use the available tools to explore the repository structure and locate the
     source and test directories.
   (2) Based on the gold patch and the original test patch, identify the most relevant
     source files and test files, and analyze how the original test verifies the fix.
3. Expand Test Coverage:
   (1) Extend the original test into a broader test suite, covering edge cases, corner
     cases, and stress conditions.
   (2) When writing tests, ensure generality and avoid overfitting assertions to the
     specific implementation of the gold patch.
      - Focus on verifying the correctness of behavior, not on matching the exact
     output form of the gold patch.
   (3) Run the tests to confirm that all pass with the gold patch applied.
   (4) If any tests fail, adjust the inputs or expected outputs until a stable and
     correct test suite is achieved.
4. Finalize Test Suite:
   (1) Consolidate both the original and extended tests into a single new test file,
     rather than providing a git diff.
```

## A.1.2. STAGE I: TEST DECOUPLING PROMPT

```
Now you must review the tests you have written:

1. Check whether there are any hard-coded assertions on error messages.
   If so, determine whether the assertion relies on keywords, fields, or formatting
     specifics introduced by the gold patch;
   if it does, replace it with a more general assertion that verifies only the
     essential error semantics rather than exact strings.

2. Check whether any test enforces strict ordering when the Issue does not require
     ordering.
   If so, revise the test to avoid relying on order.

3. Check for any other situations where the test is too closely tied to the gold patch
     implementation
   If so, adjust to avoid unnecessary coupling.
```

```
Do not create new test files again. If you believe any of the new tests you previously
    wrote may suffer from the issues above, revise them directly.
Do not install any additional libraries. If your test files import libraries that are
    not available in the current environment, replace them with alternatives that are
    available.

## Testing
After making the necessary modifications, run your tests again using:
{{test_command}}

IMPORTANT! Execute the test_command by itself. Do not combine it with any other
    commands.

If any tests fail, adjust the inputs or expected outputs until a stable and correct
    test suite is achieved.

## Submission
Before submitting, you MUST execute the following Git command to inspect the modified
    files.
You MUST use this command (or an equivalent one that only lists file names and does
    not show diffs):

```bash
git status --short
```
Based on the output of this command, you MUST remove any unnecessary or temporary
    generated files
(e.g., cache files, logs, build artifacts, or intermediate outputs).

You MUST repeat the file-listing step if files are removed, until only intended
    changes remain.

Only after completing the cleanup and when no further progress can be made,
you may issue EXACTLY the following command.

IMPORTANT:
The final submission command MUST be sent as a standalone instruction and MUST NOT be
    combined with any other instructions.

```bash
cd {{workdir}} && echo COMPLETE_TASK_AND_SUBMIT_FINAL_OUTPUT && git add -A && git diff
    --cached
```

# Workflow (You must follow this workflow):
  1. Review existing tests for hard-coded assertions, ordering assumptions, and tight
     coupling to the gold patch.
  2. (Optional) Revise the problematic tests directly to make them more general and
     robust.
  3. Run the test suite using the provided test command and fix any remaining failures.
  4. Once all tests pass and no further improvements are possible, submit the final
     changes.
```

### A.1.3. STAGE I: COVERAGE-GUIDED AUGMENTATION PROMPT

```
The test you wrote covers {{coverage_rate}} of the lines modified by the gold patch.
Now you need to update your test cases to cover these lines of code.
Do not create new test files again.
Miss line:
{{error_info}}

## Testing
After making the necessary modifications, run your tests again using:
{{test_command}}
```

```
If any tests fail, adjust the inputs or expected outputs until a stable and correct
    test suite is achieved.

## Submission
Before submitting, you MUST execute the following Git command to inspect the modified
    files.
You MUST use this command (or an equivalent one that only lists file names and does
    not show diffs):

```bash
git status --short
```
Based on the output of this command, you MUST remove any unnecessary or temporary
    generated files
(e.g., cache files, logs, build artifacts, or intermediate outputs).

You MUST repeat the file-listing step if files are removed, until only intended
    changes remain.

Only after completing the cleanup and when no further progress can be made,
you may issue EXACTLY the following command.

IMPORTANT:
The final submission command MUST be sent as a standalone instruction and MUST NOT be
    combined with any other instructions.

```bash
cd {{workdir}} && echo COMPLETE_TASK_AND_SUBMIT_FINAL_OUTPUT && git add -A && git diff
    --cached
```

# Workflow (You must follow this workflow):
    1. Identify the missed lines reported in the coverage feedback.
    2. Update existing test cases to exercise the uncovered code paths.
    3. Run the test command and verify that coverage improves and all tests pass.
    4. Iterate on test inputs or expectations until the test suite is stable.
    5. Submit the final changes.
```

### A.1.4. STAGE II: MUTANT GENERATION PROMPT

```
You are a helpful assistant that can interact multiple times with a computer shell to
    solve programming tasks.
Your response must contain exactly ONE bash code block with ONE command (or commands
    connected with && or ||).

Include a THOUGHT section before your command where you explain your reasoning process.
Format your response as shown in <format_example>.

<format_example>
THOUGHT: Your reasoning and analysis here

```bash
your_command_here
```
</format_example>

Failure to follow these rules will cause your response to be rejected.

<pr_description>
Consider the following PR description:
{{task}}
</pr_description>
```

```
<instructions>
# Task Instructions

## Overview
You are an autonomous software engineer interacting with a live repository.
Your job is to synthesize a mutation patch that:

1. Alters program behavior in subtle but semantically meaningful ways
   (NOT formatting or refactoring - but true behavior deviations)

2. Still passes ALL tests defined in **the given** Test Patch

3. Does NOT implement **the** correct fix described in **the** Gold Patch
   (Your mutation must be wrong but test-passing)

This creates a *surviving mutant* that exposes weaknesses in **the** existing tests.

There is a gold patch that is used to fix **the** issue.
--- Gold Patch ---
```
{{gold_patch}}
```

There is an test patch that is used to verify **the** fix.
---Test Patch ---
```
{{test_patch}}
```
Test execution command: {{test_command}}

IMPORTANT:
- This is an interactive process where you will think and issue ONE command, see its
    result, then think and issue your next command.
- {{workdir}}  is **the** working directory **for** all your subsequent commands.
- Test Patch and Gold Patch is already apply to **the given** repository.
- Do not run **the** full test suite, just run **the** test in **the** test patch
- Do NOT run git commit

## Hard Requirements **for** Your Mutation
1. Directly modify **the** repository source code
2. Mutation must change semantics (examples):
  - Modify branching logic
  - Alter returned values
  - Skip a step of processing
  - Incorrect edge case handling
  - Loosen a condition
3.Your mutation MUST be logically connected to **the** original issue
3. Your mutation MUST cause behavior divergence relative to **the** gold patch
  (i.e., gold_patch -> correct; your_patch -> subtly incorrect behavior)
4. After mutation, **the** repository MUST PASS all tests in **the** Test Patch
5. Absolutely forbidden:
  - Comment-only or formatting changes
  - Pure refactoring with identical semantics
  - Dead code insertion
  - Re-implementing **the** gold patch
6. Before writing your mutation, explicitly state:
  **Why this mutation is semantically wrong but can evade **the** tests.**

## Required Workflow
1. Establish baseline test status
  - Execute **the** test command on **the** current codebase
  - Record all failing test cases as baseline failures
  - These failures are considered acceptable and should NOT be fixed
  - Any additional failures introduced later must be eliminated
```

```
2. Understand the issue
   - Read PR description + gold patch + test patch
   - Identify the core expected behavior
   - Infer what aspects the test suite fails to validate or under-specifies

3. Locate files
   - Use commands to search the codebase
   - Identify where the gold patch would apply
   - Locate alternative or related code paths implementing similar logic

4. Design an adversarial mutation
   - Intentionally introduce a logically incorrect or weakened behavior
   - The mutation may preserve baseline failures
   - The mutation must NOT introduce new failing test cases

5. Iteratively refine
   - Apply the mutation and execute the tests
   - Compare failing tests against the baseline failures
   - If new test failures are observed -> adjust the mutation
   - Continue until a surviving mutant is produced with no additional failures

6. Submit the final mutation
   - Include a justification of:
     - What semantic behavior was altered or broken
     - Why the existing tests cannot catch this change
```

### A.1.5. STAGE II: RELEVANCE AND EQUIVALENCE FILTERING PROMPT

```
You are a highly experienced software developer tasked with evaluating the quality of
    a Mutation in a codebase.

You will receive the following four parts:

1. GitHub Issue Description - Background of the problem and the fix objective;
2. Gold Patch - The official or correct fix patch;
3. Test Patch - The tests added to verify the fix;
4. Mutation - A small code change introduced based on the gold patch, which has passed
    all the new tests.

Your task is to evaluate whether this Mutation is a high-quality surviving mutant.

#### Step 1: Difference Identification (Difference Analysis)

First, compare the Mutation and Gold Patch,
Summarize their differences (such as added/removed/modified logic, comments, condition
    checks, function calls, etc.).
The output should include:

 - Modified Lines (you can reference the key context)
 - Change Type (Added/Modified/Deleted)
 - A brief explanation of the semantic change or lack thereof (e.g., "only a comment
   difference," "condition branch logic adjustment")

#### Step 2: In-depth Evaluation (Structured Evaluation)

1. Relevance to the Issue (Relevance)
   - Does the Mutation modify the part of the code related to the Issue fix?
   - Does it still fall within the scope of the Issue's logic (e.g., the same function,
     path, or input conditions)?
   - If it is a mutation to a parameter, does the issue explicitly or implicitly
     specify the original value of that parameter? If not, then it is irrelevant.
   - If it's unrelated (e.g., just style or comment changes), its relevance is low.
```

```
2. Does it constitute a valid mutation? (Mutation Validity)
  - Does the **Mutation** introduce a semantic difference compared to the **Gold
    Patch**?
  - Is this change likely to alter behavior in certain scenarios and potentially cause
    a functional bug, or is it an equivalent mutation?
  - Can this difference reveal a blind spot in the test suite?
  - A high-quality surviving mutant should meet the following criteria:
    * The functionality is mostly correct, but with slight semantic deviation.
    * The test suite did not detect it.
    * It provides insight into potential test improvements.

### Output Format (Please Follow This Exactly)
Difference Summary:
  Briefly list all the logical differences between the **Gold Patch** and the
    **Mutation**.

Reasoning Analysis:
  Relevance to the Issue:
  Mutation Validity (Does it truly constitute a mutation?):

Final Evaluation: Please output the final judgment in the following format, placed
    inside the <Answer></Answer> tag:
<Answer>
Relevance: Yes/No
Mutation Validity: Yes/No
</Answer>

### Input Format

```
## Issue
{{issue}}

## Test Patch
{{test_patch}}

## Gold Patch
{{gold_patch}}

## Mutation
{{mutation}}
```
```

### A.1.6. STAGE II: PROMPT FOR MUTANT-GUIDED TEST AUGMENTATION OF NON-EQUIVALENT MUTANTS

```
You are a helpful assistant that can interact multiple times with a computer shell to
    solve programming tasks.
Your response must contain exactly ONE bash code block with ONE command (or commands
    connected with && or ||).

Include a THOUGHT section before your command where you explain your reasoning process.
Format your response as shown in <format_example>.

<format_example>
THOUGHT: Your reasoning and analysis here

```bash
your_command_here
```

</format_example>
```

Failure to follow these rules will cause your response to be rejected.

<pr_description>
Consider **the** following PR description:
{{task}}
</pr_description>

<instructions>
# Task Instructions

## Overview
You're a software engineer interacting continuously with a computer by submitting
    commands.
Your task is to directly modify and enhance **the** current test code to ensure that logic
    deviations can be detected.

You are **given** three code snippets related to a specific GitHub issue:
There is a gold patch that is used to fix **the** issue.
1. Gold Patch – **the** correct fix **for the** issue.
   ---
   {{gold_patch}}
   ---

2. Mutation Patch – a mutated version of **the** gold patch that still passes **the** existing
    tests (a surviving mutant).
   ---
   {{mutation_patch}}
   ---

3. Mutation thinking derived from mutations in **the** Gold Patch.
   ---
   {{mutation_thinking}}
   ---

4. Test Patch – **the** existing test code that was originally used to verify **the** gold
    patch.
   ---
   {{test_patch}}
   ---
Test execution in this repository: {{test_command}}

Your goal is to **enhance **the given** test code** (from **the** Test Patch) so that:
- **The** **Gold Patch** still passes all tests (it represents **the** correct behavior).
- **The** **Mutation Patch** fails at least one test (it represents an incorrect but
    surviving variant).

Requirements **for the** new test:
- Modify only **the** test code.
- Add minimal but **targeted** assertions or test cases that can expose **the** behavioral
    difference between **the** Gold Patch and **the** Mutation Patch.

IMPORTANT:
- This is an interactive process where you will think and issue ONE command, see its
    result, then think and issue your next command.
- {{workdir}}  is **the** working directory **for** all your subsequent commands.
- You will be working with two repositories: **Gold** and **Mutated**, which have
    respectively applied **the** Gold Patch and **the** Mutation Patch.
- **The** Test Patch has already been applied to both repositories.
- Do not run **the** full test suite, just run **the** test in **the** test patch
- Do not change **the** test command, only use **the given** test command
- When writing commands, always specify which repository you are operating on. **For**
    example: `<env>Gold</env> cd testbed && echo 1` or `<env>Mutated</env> cd testbed
    && echo 1`.

```
- If you want to run the command in both repositories, you can use the
  `<env>All</env>` tag. For example: `<env>All</env> cd testbed && echo 1` will run
  `cd testbed && echo 1` in both Gold and Mutated repositories.
- If you use the <env>All</env>tag, do not include any other environment-specific tags
  afterward, as it will cause a parsing error.

## Recommended Workflow
1. Understand the Problem:
   (1) Begin by carefully reading the user's problem description to fully grasp the
     issue.
   (2) Identify the core components and expected behavior.
2. Explore and Locate:
   (1) Use the available tools to explore the repository structure and locate the
     source and test directories.
   (2) Based on the gold patch and the test patch, identify the most relevant source
     files and test files
   (3) Identify the diffence between the gold patch and the mutation patch
3. Modify the test file and debug:
   (1) Modify and enhance the current test code in test_patch
   (2) Debug the test code to ensure it passes the gold patch and fails the mutation
     patch
   (3) If any tests fail with gold patch, adjust the inputs or expected outputs until a
     stable and correct test suite is achieved.
   (4) Ensure that the newly added test cases are also general, rather than being
     gold-patch-specific.
4. Submit
   (1) Before you submit, run the given tests command to ensure that all tests pass
     with the gold patch applied and fail with the mutation patch applied.
```

A.1.7. STAGE II: PROMPT FOR MUTANT-GUIDED TEST AUGMENTATION OF EQUIVALENT MUTANTS

```
<pr_description>
Consider the following PR description:
{{task}}
</pr_description>

<instructions>
# Task Instructions

## Overview
You're a software engineer interacting continuously with a computer by submitting
   commands.
Your task is to directly modify the current test code to ensure that the equivalent
   mutation (equ_mutation) can pass.

You are given three code snippets related to a specific GitHub issue:
1. Gold Patch – the correct fix for the issue.
   ---
   {{gold_patch}}
   ---

2. Mutation Patch – a equivalent mutated version of the gold patch that fails the
   existing tests.
   ---
   {{mutation_patch}}
   ---

3. Test Patch – the existing test code that was originally used to verify the gold
   patch.
   ---
   {{test_patch}}
   ---
```

```
Test execution in this repository: {{test_command}}

Your goal is to **modify the given test code** (from the Test Patch) so that:
- The **Gold Patch** and **Mutation Patch** passes all tests (it represents the
    correct behavior).

Requirements for the new test:
- Modify only the test code.

IMPORTANT:
- This is an interactive process where you will think and issue ONE command, see its
    result, then think and issue your next command.
- /testbed  is the working directory for all your subsequent commands.
- You will be working with two repositories: **Gold** and **Mutated**, which have
    respectively applied the Gold Patch and the Mutation Patch.
- The Test Patch has already been applied to both repositories.
- Do not run the full test suite, just run the test in the test patch
- Do not change the test command, only use the given test command
- When writing commands, always specify which repository you are operating on. For
    example: `<env>Gold</env> cd testbed && echo 1` or `<env>Mutated</env> cd testbed
    && echo 1`.
- If you want to run the command in both repositories, you can use the
    `<env>All</env>` tag. For example: `<env>All</env> cd testbed && echo 1` will run
    `cd testbed && echo 1` in both Gold and Mutated repositories.
- If you use the <env>All</env>tag, do not include any other environment-specific tags
    afterward, as it will cause a parsing error.

## Recommended Workflow
0. Check if it is an equivalent mutation:
   (1) If it is not an equivalent mutation, submit directly.
1. Understand the Problem:
   (1) Begin by carefully reading the user's problem description to fully grasp the
     issue.
   (2) Identify the core components and expected behavior.
2. Explore and Locate:
   (1) Use the available tools to explore the repository structure and locate the
     source and test directories.
   (2) Based on the gold patch and the test patch, identify the most relevant source
     files and test files
   (3) Identify the parts of the Test Patch that cause the mutation patch to fail.
3. Modify the test file and debug:
   (1) Modify and enhance the current test code in test_patch
   (2) Debug the test code to ensure it passes the gold patch and the mutation patch
   (3) If any tests fail with gold patch and the mutation patch, adjust the inputs or
     expected outputs until a stable and correct test suite is achieved.
4. Submit
   (1) Before you submit, run the given tests command to ensure that all tests pass
     with the gold patch applied and mutation patch applied.
```

## A.2. Overfit Tests: Definition and Case Studies

### A.2.1. FORMAL DEFINITION OF OVERFIT TESTS

A test $t$ is overfit with respect to gold patch $p_g$ if and only if there exists a semantically correct alternative patch $p'$ (one that fully resolves the reported issue) such that $t(p_g)$ passes but $t(p')$ fails. In other words, $t$ encodes an implementation choice specific to $p_g$ rather than a behavioral requirement of the issue. Because exhaustively enumerating all valid alternative patches is infeasible, we operationalize this definition through conservative proxies: we flag tests whose assertions bind to details non-essential to issue resolution, including exact error-message wording, helper-call structure, internal variable naming, specific numeric constants, logging format, return-value representation, and execution order of side effects. Flagged tests are then generalized or discarded as described in Section 3.1.2.

A.2.2. CASE STUDY

**Instance.** astropy__astropy-13033

**Issue.** When a required column other than `time` is removed from a `TimeSeries` object, the raised exception is misleading. Instead of indicating that a required column is missing, the error message claims that the first column is invalid, sometimes comparing identical values (e.g., `'time'` vs. `'time'`), which confuses users.

**Gold Fix.** Listing 1 shows the gold patch that corrects the exception message. Rather than hard-coding the first required column, the fix reports the full list of required columns and the actual columns found, producing a semantically accurate and less confusing error message.

*Listing 1.* Gold patch: corrected error message for missing required columns

```
@@ -76,9 +83,10 @@ def _check_required_columns(self):

            elif self.colnames[:len(required_columns)] != required_columns:

-               raise ValueError("{} object is invalid - expected '{}' "
-                                "as the first column{} but found '{}'"
-                                .format(self.__class__.__name__, required_columns[0],
    ↪ plural, self.colnames[0]))
+               raise ValueError("{} object is invalid - expected {} "
+                                "as the first column{} but found {}"
+                                .format(self.__class__.__name__, as_scalar_or_list_str(
    ↪ required_columns),
+                                      plural, as_scalar_or_list_str(self.colnames[:
    ↪ len(required_columns)])))
```

**Original Test Patch.** Listing 2 presents the original test added alongside the fix. This test validates the new behavior by asserting the exact error message string, tightly coupling the test to a specific formatting of the exception message.

*Listing 2.* Original test patch asserting exact error message

```
diff --git a/astropy/timeseries/tests/test_sampled.py b/astropy/timeseries/tests/
    ↪ test_sampled.py
--- a/astropy/timeseries/tests/test_sampled.py
+++ b/astropy/timeseries/tests/test_sampled.py
@@ -395,6 +395,14 @@ def test_required_columns():
     assert exc.value.args[0] == ("TimeSeries object is invalid - expected "
                                  "'time' as the first column but found 'banana'")

+    # https://github.com/astropy/astropy/issues/13009
+    ts_2cols_required = ts.copy()
+    ts_2cols_required._required_columns = ['time', 'a']
+    with pytest.raises(ValueError) as exc:
+        ts_2cols_required.remove_column('a')
+    assert exc.value.args[0] == ("TimeSeries object is invalid - expected "
+                                 "['time', 'a'] as the first columns but found ['time', '
    ↪ b']")
+

 @pytest.mark.parametrize('cls', [BoxLeastSquares, LombScargle])
 def test_periodogram(cls):
```

**Initial Generated Test (Overfitted).** Listing 3 shows an automatically generated test that further overfits to a specific error message format. Although functionally correct, the test assumes a particular phrasing and column representation, making it fragile to legitimate refactorings of the error message.

*Listing 3.* Overfitted generated test relying on exact error message

```
+def test_relax_mode_invalid_first_column_with_multiple_required():
+    # Empty initialization triggers relax mode
+    ts = TimeSeries()
+    # Set multiple required columns
+    ts._required_columns = ['time', 'a']
+    # Add wrong first column when relax=True should give single-column message
+    with pytest.raises(ValueError) as exc:
+        ts.add_column(Column([1, 2, 3], name='flux'), index=0)
+    assert exc.value.args[0] == ("TimeSeries object is invalid - expected "
+                                 "'time' as the first column but found 'flux'")
```

**Decoupled Test.** Listing 4 presents the improved, decoupled version of the test. Instead of asserting the full error string, the test checks for key semantic elements (e.g., the presence of "expected", "found", and the relevant column names). This approach preserves test robustness while still validating the intended behavior.

*Listing 4.* Decoupled test asserting semantic properties of the error message

```
+def test_relax_mode_invalid_first_column_with_multiple_required():
+    ts = TimeSeries()
+    ts._required_columns = ['time', 'a']
+    with pytest.raises(ValueError) as exc:
+        ts.add_column(Column([1, 2, 3], name='flux'), index=0)
+    msg = str(exc.value)
+    assert "expected" in msg and "found" in msg
+    assert "first column" in msg
+    assert "'time'" in msg and "'flux'" in msg
```

## A.3. Ablation: Effect of Test Decoupling

To isolate the contribution of the test-decoupling module (Section 3.1.2), we run the full SWE-ABS pipeline with and without decoupling on a randomly sampled 100-instance subset of SWE-Bench Verified (seed=10). Table 6 reports the result.

*Table 6.* Effect of removing the test-decoupling module on a 100-instance subset of SWE-Bench Verified.

| Setting | Strengthened | Avg Drop | Overfit Cases |
|---|---|---|---|
| w/o decoupling | 59/100 | 24.40% | 17 |
| w/ decoupling | 57/100 | 20.50% | 9 |

Removing decoupling yields marginally more strengthened instances (59 vs. 57) and a higher apparent resolution-rate drop (24.40% vs. 20.50%), but the number of overfit cases nearly doubles ($9 \rightarrow 17$). This indicates that the additional rejections obtained without decoupling largely reflect overfitted tests rather than genuine strengthening: agents are penalized for valid alternative implementations rather than for incorrect ones. The decoupling module thus trades a small amount of apparent rejection power for a substantial reduction in overfitting, which is consistent with the conservative false-negative rate of 10.6% reported on the full 500-instance benchmark (Section 4.4.1).

## A.4. Detailed Experimental Implementation

### A.4.1. HYPERPARAMETERS

- LLM temperature: 1.0 (default)

- Max test generation attempts: 3

- Max mutants generated per instance: 2

- Max test augmentation attempts per adversarial mutant: 3

- Test execution timeout: 120 seconds

- Max dependency graph depth: 5 hops

### A.4.2. PROGRAM SLICING IMPLEMENTATION

We perform intraprocedural program slicing to identify code regions that are data-flow or control-flow dependent on a given patch. We use Tree-sitter (Tree-sitter, 2026) to parse source files across multiple languages, including Python, JavaScript, Go, and TypeScript.

1. **AST Parsing and Scope Identification**: Parse modified source files into abstract syntax trees and build a line-to-scope mapping that identifies whether each line belongs to a function, class, or global scope. This enables scope-limited slicing to prevent excessive propagation across unrelated code regions.

2. **Executable Line Extraction**: Identify executable statements (assignments, returns, function/class definitions, etc.) and filter out non-executable lines such as comments, docstrings, and blank lines. For multi-line statements (e.g., function signatures, method calls), we map modified lines within the statement to its starting line. We denote the set of all executable lines as $L_{\text{executable}}$.

3. **Global Modification Filtering**: Filter out semantically insignificant global-level modifications such as import statements and simple variable assignments, retaining only modifications with substantial semantic impact (e.g., function-/class definitions, control flow statements).

4. **Def-Use Analysis**: Construct data dependency information by analyzing variable definitions (assignments) and uses (loads) at each line using an AST visitor pattern.

5. **Bidirectional k-Hop Slicing**:

- **Forward slicing**: Starting from modified lines, propagate through def-use chains to find lines that use variables defined by the patch (def $\rightarrow$ use). We denote the forward slice as $L_{\text{fwd}}$.

- **Backward slicing**: Propagate through use-def chains to find lines that define variables used by the patch (use $\rightarrow$ def). We denote the backward slice as $L_{\text{bwd}}$.

- Use $k$-hop propagation (typically $k = 1$ or $k = 5$) to limit slice size while capturing relevant dependencies.

- Optionally restrict propagation to the same scope (function/class/global) as modified lines to avoid excessive expansion.

6. **Slice Union**: Combine forward and backward slices, then intersect with executable lines to obtain the final patch-relevant lines $L_{\text{rel}} = (L_{\text{fwd}} \cup L_{\text{bwd}}) \cap L_{\text{executable}}$.

This design balances precision and scalability: scope-limited slicing prevents uncontrolled expansion, while bidirectional propagation captures both upstream dependencies (what the patch depends on) and downstream impacts (what depends on the patch).

### A.4.3. COVERAGE CALCULATION

To measure the effectiveness of coverage-driven augmentation (Stage I), we compute the percentage of patch-relevant lines $L_{\text{rel}}$ executed by a given test suite using:

$$\text{Coverage}(T, L_{\text{rel}}) = \frac{|L_{\text{exec}} \cap L_{\text{rel}}|}{|L_{\text{rel}}|},$$

where $L_{\text{exec}}$ denotes the set of executed lines and $L_{\text{rel}}$ denotes patch-relevant lines from program slicing. This formula applies uniformly across both benchmarks; the key difference lies in how we collect $L_{\text{exec}}$ for different programming languages and repository configurations.

**SWE-Bench (Python).** For SWE-Bench instances, we use `trace.py` from the SWT-Bench (Mündler et al., 2024) repository to collect line-level execution traces.

**SWE-Bench Pro (Multi-Language).** SWE-Bench Pro introduces instances in Python, JavaScript, Go, and TypeScript, requiring language-specific coverage tools. We implement a unified coverage parsing framework that standardizes output from heterogeneous tools into a common representation:

**Python**: We use `coverage.py` (Batchelder, 2026) to collect line-level execution traces.

**Go**: We use Go's built-in coverage profiler (`go test -coverprofile`) (Pike, 2013), which produces line-range coverage data that we parse to extract executed lines.

**JavaScript**: We employ Istanbul/nyc (Istanbuljs, 2024) to collect statement-level coverage, from which we extract executed line numbers.

**TypeScript**: We use Node.js V8 coverage (V8 Developers, 2017), which provides byte-offset-based coverage that we convert to line numbers through exact mapping.

All language-specific parsers output a standardized `CoverageResult` structure containing executed and missing lines per file. Once $L_{\text{exec}}$ is extracted for each language, we apply the same coverage formula uniformly across all instances.

### A.4.4. SPEARMAN CALCULATION

To quantify the stability of leaderboard rankings under test augmentation, we compute the Spearman rank correlation coefficient $\rho$ between the original and augmented agent rankings.

**Mathematical Definition.** Given $n$ agents with original ranks $r_i$ and augmented ranks $r_i'$ (where $i = 1, \ldots, n$), Spearman's $\rho$ is defined as:

$$\rho = 1 - \frac{6 \sum_{i=1}^{n} d_i^2}{n(n^2 - 1)},$$

where $d_i = r_i - r'_i$ is the rank difference for agent $i$.

This formula measures the monotonic relationship between two rankings. A value of $\rho = 1$ indicates perfect rank agreement (identical orderings), $\rho = 0$ indicates no correlation, and $\rho = -1$ indicates perfect rank reversal. In our evaluation, lower $\rho$ values signify greater leaderboard instability induced by test strengthening.

**Implementation.** We compute Spearman's $\rho$ using the `spearmanr` function from the `scipy.stats` module (Virtanen et al., 2020):

**Interpretation in Our Context.** For RQ1 on SWE-Bench Verified, we evaluate 30 agents whose leaderboard positions may change under augmented tests. The Spearman $\rho$ of 0.79 under SWE-ABS (compared to 0.98 under UTBoost) indicates substantial ranking instability, revealing that the original test suites provide unreliable assessments of agent capabilities.

## A.5. Mutation Cases and Evaluation

### A.5.1. EQUIVALENT MUTATION CASE

**Instance ID.** matplotlib_matplotlib-20826

**Issue** In figures with shared axes created by `plt.subplots(..., sharex=True, sharey=True)`, calling `ax.clear()` resets axis properties related to tick and tick-label visibility. As a result, tick labels that should be hidden for shared axes (e.g. inner subplots) become visible, and extra ticks appear on the top and right spines.

This is a regression introduced in matplotlib 3.4.2. While the shared-axis linkage (e.g. synchronized limits) is preserved after `ax.clear()`, the visual state enforced by shared axes is lost. In matplotlib 3.4.1, `ax.clear()` preserved the expected tick visibility for shared axes.

**Gold Patch.** Listing 5 presents the gold patch that fixes the regression in `Axis.clear()`. Instead of fully resetting the major and minor tick keyword dictionaries, the patch explicitly restores the `gridOn` state from the global `rcParams`. This preserves tick and tick-label visibility semantics expected for shared axes, preventing hidden ticks from being unintentionally re-enabled after `ax.clear()`.

*Listing 5.* Gold patch: preserve tick visibility when clearing shared axes

```
diff --git a/lib/matplotlib/axis.py b/lib/matplotlib/axis.py
--- a/lib/matplotlib/axis.py
+++ b/lib/matplotlib/axis.py
@@ -806,8 +806,13 @@ def clear(self):
         # Clear the callback registry for this axis, or it may "leak"
         self.callbacks = cbook.CallbackRegistry()

-        self._reset_major_tick_kw()
-        self._reset_minor_tick_kw()
+        # whether the grids are on
+        self._major_tick_kw['gridOn'] = (
+                mpl.rcParams['axes.grid'] and
+                mpl.rcParams['axes.grid.which'] in ('both', 'major'))
+        self._minor_tick_kw['gridOn'] = (
+                mpl.rcParams['axes.grid'] and
+                mpl.rcParams['axes.grid.which'] in ('both', 'minor'))
         self.reset_ticks()

         self.converter = None
```

**Mutation Patch.** Listing 6 presents a mutation patch that is semantically equivalent to the gold patch. The only difference lies in the omission of a descriptive comment, while the executable statements modifying the `gridOn` flags remain identical. As a result, both patches exhibit the same runtime behavior.

*Listing 6.* Mutation patch: comment-only variant of the gold patch

```
diff --git a/lib/matplotlib/axis.py b/lib/matplotlib/axis.py
```

```
index 887cf97ef1..7feca94bd9 100644
--- a/lib/matplotlib/axis.py
+++ b/lib/matplotlib/axis.py
@@ -806,8 +806,12 @@ class Axis(martist.Artist):
         # Clear the callback registry for this axis, or it may "leak"
        self.callbacks = cbook.CallbackRegistry()

-       self._reset_major_tick_kw()
-       self._reset_minor_tick_kw()
+       self._major_tick_kw['gridOn'] = (
+               mpl.rcParams['axes.grid'] and
+               mpl.rcParams['axes.grid.which'] in ('both', 'major'))
+       self._minor_tick_kw['gridOn'] = (
+               mpl.rcParams['axes.grid'] and
+               mpl.rcParams['axes.grid.which'] in ('both', 'minor'))
        self.reset_ticks()

        self.converter = None
```

**Mutation Rationale (from model).** I introduced a semantically wrong mutation in Axis.clear(): instead of honoring rcParams['axes.grid.which'] to separately control major vs minor grid activation, I set both major and minor gridOn flags solely based on rcParams['axes.grid']. This breaks the intended distinction and is logically connected to the original issue area (grid/ticks handling on clear).

**Judge.** Although the model articulated a mutation rationale that intended to introduce a semantic change in the handling of major and minor grid activation, the actual mutation does not implement this logic. In the produced patch, the executable statements are identical to those of the gold patch, differing only in the presence of comments. As a result, the mutation does not alter runtime behavior and constitutes an equivalent mutant. This discrepancy suggests a mismatch between the model's stated intent and the realized code change, likely due to hallucination or imprecise code generation.

A.5.2. NON-EQUIVALENT MUTATION CASE

**Instance.** astropy__astropy-14365

**Issue.** The ascii.qdp module assumes that commands in QDP files are upper case (e.g., "READ SERR 1 2"), but QDP itself is case-insensitive and accepts lowercase commands (e.g., "read serr 1 2"). Since many QDP files are created by hand, the parser should support case-insensitive commands and the special value "NO" (indicating masked data) in any case variation.

**Gold Patch (Production Fix).** Listing 7 shows the gold patch that fully addresses the issue. The fix makes command parsing case-insensitive by compiling the regular expression with re.IGNORECASE, and it also normalizes data tokens using v.upper() == "NO", ensuring that all case variants of NO are handled uniformly.

*Listing 7.* Gold patch: fully case-insensitive QDP parsing

```
@@ -68,7 +68,7 @@ def _line_type(line, delimiter=None):
    _new_re = rf"NO({sep}NO)+"
    _data_re = rf"({_decimal_re}|NO|[-+]?nan)({sep}({_decimal_re}|NO|[-+]?nan))*)"
    _type_re = rf"^\s*((?P<command>{_command_re})|(?P<new>{_new_re})|(?P<data>{_data_re})
  ↪ ?\s*(\!(?P<comment>.*))?\s*$"
-   _line_type_re = re.compile(_type_re)
+   _line_type_re = re.compile(_type_re, re.IGNORECASE)
    line = line.strip()
    if not line:
        return "comment"
@@ -306,7 +306,7 @@ def _get_tables_from_qdp_file(qdp_file, input_colnames=None, delimiter
  ↪ =None):

            values = []
            for v in line.split(delimiter):
-               if v == "NO":
```

```
+                    if v.upper() == "NO":
                         values.append(np.ma.masked)
                     else:
                         # Understand if number is int or float
```

**Mutation Patch.** Listing 8 shows a mutation patch that appears to fix the issue but is not semantically equivalent to the gold patch. While it correctly enables case-insensitive command matching via `re.IGNORECASE`, it only handles two specific variants of the masked value (`"NO"` and `"no"`). Other valid case combinations (e.g., `"No"`, `"nO"`) are still rejected, making this mutation incomplete.

*Listing 8.* Mutation patch: incomplete handling of case-insensitive masked values

```
@@ -68,7 +68,7 @@ def _line_type(line, delimiter=None):
    _new_re = rf"NO({sep}NO)+"
    _data_re = rf"({_decimal_re}|NO|[-+]?nan)({sep}({_decimal_re}|NO|[-+]?nan))*)"
    _type_re = rf"^\s*((?P<command>{_command_re})|(?P<new>{_new_re})|(?P<data>{_data_re})
    ↪ ?\s*(\!(?P<comment>.*))?\s*$"
-    _line_type_re = re.compile(_type_re)
+    _line_type_re = re.compile(_type_re, re.IGNORECASE)
    line = line.strip()
    if not line:
        return "comment"
@@ -306,7 +306,7 @@ def _get_tables_from_qdp_file(qdp_file, input_colnames=None, delimiter
    ↪ =None):

            values = []
            for v in line.split(delimiter):
-                if v == "NO":
+                if v == "NO" or v == "no":
                     values.append(np.ma.masked)
                 else:
                     # Understand if number is int or float
```

**Judge.** The mutation makes the same change to the regex compilation (adding `re.IGNORECASE`), but differs in how it handles the "NO" keyword for masked values. The gold patch uses `v.upper() == "NO"`, which correctly handles all case variations ("NO", "no", "No", "nO", etc.). The mutation only checks for `v == "NO" or v == "no"`, which only handles all-uppercase and all-lowercase cases. The mutation passes all existing tests because the test suite only uses "NO" and "no", but it fails to handle mixed-case variations like "No" or "nO". This is a **non-equivalent mutation** that introduces a real bug not detected by current tests.

A.5.3. HUMAN EVALUATION OF MUTATION CLASSIFICATION

To validate the accuracy of our LLM-based mutation judge, we randomly sampled 100 mutants from our dataset and manually annotated whether each mutation was equivalent or non-equivalent. We then compared these human annotations with the LLM judge's classifications. The results show that the LLM judge achieved a 98% agreement rate with human judgments, demonstrating high reliability in distinguishing between equivalent mutations (which preserve the original behavior) and non-equivalent mutations (which introduce meaningful behavioral changes that expose test suite weaknesses). This high agreement rate confirms that our automated mutation classification approach is robust and can be trusted for large-scale mutation analysis.

**Quantitative alignment with real agent errors.** To directly assess whether our adversarial mutants resemble realistic agent mistakes, we randomly sampled 100 mutants and classified each under the same taxonomy used for rejected agent patches in Table 5 (Section 4.4). Table 7 shows that the two dominant error families align across mutants and agent patches: logic errors are most frequent in both (∼47%) and incomplete fixes second (∼25–29%). This alignment indicates that SWE-ABS mutants are not synthetic edge cases but plausible failure modes that current agents actually produce.

*Table 7.* Error-type distribution of adversarial mutants vs. rejected agent patches under the same taxonomy ($n = 100$ each).

| Error Category | Adversarial Mutants | Agent Patches |
| --- | :---: | :---: |
| Logic errors | 47% | 46% |
| Incomplete fixes | 25% | 29% |
| Type mismatches | 5% | 13% |
| Boundary violations | 23% | 12% |

## A.6. Top-30 Agents Performance in SWE-ABS

*Table 8.* Full leaderboard results for Top-30 agents on SWE-Bench Verified under SWE-ABS augmented test suites.

| Date | Model | Orig. (%) | SWE-ABS (%) | Drop (%) | Rank (Old→New) |
|---|---|---|---|---|---|
| 20250928 | TRAE + Doubao-Seed-Code | 78.80 | 62.20 | 16.60 | 1→5 |
| 20251120 | live-SWE-agent + Gemini 3 Pro Preview (2025-11-18) | 77.40 | 65.40 | 12.00 | 2→1 |
| 20250902 | Atlassian Rovo Dev (2025-09-02) | 76.80 | 61.20 | 15.60 | 3→8 |
| 20250804 | EPAM AI/Run Developer Agent v20250719 + Claude 4 Sonnet | 76.80 | 63.80 | 13.00 | 4→3 |
| 20250819 | ACoder | 76.40 | 61.80 | 14.60 | 5→6 |
| 20250901 | Warp | 75.80 | 62.40 | 13.40 | 6→4 |
| 20250612 | TRAE | 75.20 | 58.00 | 17.20 | 7→18 |
| 20251103 | Sonar Foundation Agent + Claude 4.5 Sonnet | 75.00 | 64.20 | 10.80 | 8→2 |
| 20250731 | Harness AI | 74.80 | 60.40 | 14.40 | 9→13 |
| 20250915 | JoyCode | 74.60 | 61.20 | 13.40 | 10→9 |
| 20250720 | Lingxi-v1.5_claude-4-sonnet-20250514 | 74.60 | 60.60 | 14.00 | 11→12 |
| 20250603 | Refact.ai Agent | 74.40 | 58.40 | 16.00 | 12→16 |
| 20251015 | Prometheus-v1.2.1 + GPT-5 | 74.40 | 58.40 | 16.00 | 13→17 |
| 20251124 | mini-SWE-agent + Claude 4.5 Opus medium (20251101) | 74.40 | 58.00 | 16.40 | 14→19 |
| 20251118 | mini-SWE-agent + Gemini 3 Pro Preview (2025-11-18) | 74.20 | 56.80 | 17.40 | 15→22 |
| 20251103 | Salesforce AI Research SAGE (Open-Hands) | 74.00 | 61.60 | 12.40 | 16→7 |
| 20250522 | Tools + Claude 4 Opus (2025-05-22) | 73.20 | 58.40 | 14.80 | 17→15 |
| 20251021 | Salesforce AI Research SAGE (bash-only) | 73.20 | 61.20 | 12.00 | 18→10 |
| 20250522 | Tools + Claude 4 Sonnet (2025-05-22) | 72.40 | 56.40 | 16.00 | 19→24 |
| 20250807 | OpenHands + GPT-5 | 71.80 | 61.20 | 10.60 | 20→11 |
| 20251211 | mini-SWE-agent + GPT-5.2 (2025-12-11) (high reasoning) | 71.80 | 56.80 | 15.00 | 21→23 |
| 20250715 | Qodo Command | 71.20 | 57.60 | 13.60 | 22→20 |
| 20251014 | Lingxi v1.5 x Kimi K2 | 71.20 | 57.60 | 13.60 | 23→21 |
| 20250929 | Prometheus-v1.2 + GPT-5 | 71.20 | 56.00 | 15.20 | 24→26 |
| 20250710 | Bloop | 71.20 | 59.60 | 11.60 | 25→14 |
| 20250623 | Warp | 71.00 | 52.60 | 18.40 | 26→30 |
| 20250611 | Moatless Tools + Claude 4 Sonnet | 70.80 | 56.20 | 14.60 | 27→25 |
| 20250519 | TRAE | 70.60 | 54.20 | 16.40 | 28→29 |
| 20250929 | mini-SWE-agent + Claude 4.5 Sonnet (20250929) | 70.60 | 54.40 | 16.20 | 29→28 |
| 20250515 | Refact.ai Agent | 70.40 | 54.80 | 15.60 | 30→27 |

## A.7. Bash Only Agents Performance in SWE-ABS

*Table 9.* Full leaderboard results for Bash-only agents on SWE-Bench Verified under SWE-ABS augmented test suites.

| Date | Model | Orig. (%) | SWE-ABS (%) | Drop (%) | Rank (Old→New) |
|---|---|---|---|---|---|
| 20251124 | Claude 4.5 Opus medium (20251101) | 74.40 | 58.00 | 16.40 | 1→1 |
| 20251118 | Gemini 3 Pro Preview (2025-11-18) | 74.20 | 56.80 | 17.40 | 2→3 |
| 20251211 | GPT-5.2 (2025-12-11) (high reasoning) | 71.80 | 56.80 | 15.00 | 3→2 |
| 20250929 | Claude 4.5 Sonnet (20250929) | 70.60 | 54.40 | 16.20 | 4→4 |
| 20251211 | GPT-5.2 (2025-12-11) | 69.00 | 51.60 | 17.40 | 5→5 |
| 20250802 | Claude 4 Opus (20250514) | 67.60 | 50.60 | 17.00 | 6→6 |
| 20251124 | GPT-5.1-codex (medium reasoning) | 66.00 | 49.00 | 17.00 | 7→7 |
| 20251120 | GPT-5.1 (2025-11-13) (medium reasoning) | 66.00 | 47.60 | 18.40 | 8→9 |
| 20250807 | GPT-5 (2025-08-07) (medium reasoning) | 65.00 | 48.60 | 16.40 | 9→8 |
| 20250726 | Claude 4 Sonnet (20250514) | 64.80 | 45.80 | 19.00 | 10→10 |
| 20251210 | Kimi K2 Thinking | 63.40 | 45.00 | 18.40 | 11→11 |
| 20251124 | Minimax M2 | 61.00 | 44.40 | 16.60 | 12→13 |
| 20251201 | DeepSeek V3.2 Reasoner | 60.00 | 43.40 | 16.60 | 13→15 |
| 20250807 | GPT-5 mini (2025-08-07) (medium reasoning) | 59.80 | 44.60 | 15.20 | 14→12 |
| 20250726 | o3 (2025-04-16) | 58.40 | 43.80 | 14.60 | 15→14 |
| 20251209 | Devstral small (2512) | 56.40 | 40.80 | 15.60 | 16→16 |
| 20251201 | GLM-4.6 (T=1) | 55.40 | 40.20 | 15.20 | 17→19 |
| 20250822 | GLM-4.5 (2025-08-22) | 54.20 | 38.20 | 16.00 | 18→20 |
| 20251209 | Devstral (2512) | 53.80 | 40.60 | 13.20 | 19→18 |
| 20250726 | Gemini 2.5 Pro (2025-05-06) | 53.60 | 40.80 | 12.80 | 20→17 |
| 20250807 | GPT-5 nano (2025-08-07) (medium reasoning) | 34.80 | 22.80 | 12.00 | 21→21 |
| 20250807 | gpt-oss-120b | 26.00 | 20.20 | 5.80 | 22→22 |
| 20250726 | Gemini 2.5 Flash (2025-04-17) | 24.60 | 18.80 | 5.80 | 23→23 |
| 20250720 | GPT-4o (2024-11-20) | 23.20 | 16.80 | 6.40 | 24→24 |
| 20250720 | Llama 4 Maverick Instruct | 21.00 | 15.80 | 5.20 | 25→25 |
| 20250726 | Gemini 2.0 flash | 13.20 | 9.40 | 3.80 | 26→26 |
| 20250720 | Claude 3.7 Sonnet (20250219) | 10.20 | 7.20 | 3.00 | 27→27 |
| 20250803 | Qwen2.5-Coder 32B Instruct | 9.00 | 6.60 | 2.40 | 28→28 |
| 20250720 | Llama 4 Scout Instruct | 8.00 | 5.60 | 2.40 | 29→29 |

## A.8. Comparison with PatchDiff

Table 10 compares SWE-ABS against the concurrent PatchDiff approach (Wang et al., 2025) on the three agents reported by PatchDiff. PatchDiff takes both the gold patch and a specific agent submission as input, and generates differential tests targeted at that particular submission; the produced tests are therefore tied to one agent patch and must be regenerated for each new submission. SWE-ABS instead generates a single reusable test suite from the gold patch alone, before any agent submission is observed, and reuses it across all evaluated agents. Despite this weaker input regime, SWE-ABS attains a comparable resolution-rate drop (average $-14.13$ pp vs. $-17.33$ pp).

*Table 10.* Resolution-rate drop (in percentage points) on SWE-Bench Verified (500 instances).

| Agent | PatchDiff Drop | SWE-ABS Drop |
|---|---|---|
| CodeStory | $-18.20$ | $-14.20$ |
| LearnByInteract | $-19.40$ | $-14.00$ |
| OpenHands | $-14.40$ | $-14.20$ |
| **Average** | $\mathbf{-17.33}$ | $\mathbf{-14.13}$ |

## A.9. Additional Results

*Table 11.* SWE-Bench Verified strengthening rate by repository. Smaller repositories with fewer instances (e.g., mwaskom/seaborn, pallets/flask) achieve 100% strengthening, while larger repositories such as django/django and sympy/sympy show moderate strengthening rates (around 40%).

| Repository | Instances | Strengthened | Rate |
|---|---|---|---|
| django/django | 231 | 102 | 44.2% |
| sympy/sympy | 75 | 34 | 45.3% |
| sphinx-doc/sphinx | 44 | 35 | 79.5% |
| matplotlib/matplotlib | 34 | 22 | 64.7% |
| scikit-learn/scikit-learn | 32 | 9 | 28.1% |
| astropy/astropy | 22 | 18 | 81.8% |
| pydata/xarray | 22 | 9 | 40.9% |
| pytest-dev/pytest | 19 | 9 | 47.4% |
| pylint-dev/pylint | 10 | 5 | 50.0% |
| psf/requests | 8 | 5 | 62.5% |
| mwaskom/seaborn | 2 | 2 | 100.0% |
| pallets/flask | 1 | 1 | 100.0% |

*Table 12.* SWE-Bench Pro strengthening rate by programming language. Python exhibits the highest strengthening rate (75.38%), while TypeScript shows the lowest rate (18.18%).

| Language | Instances | Strengthened | Rate |
|---|---|---|---|
| Python | 65 | 49 | 75.38% |
| Go | 48 | 30 | 62.50% |
| JavaScript | 26 | 16 | 61.54% |
| TypeScript | 11 | 2 | 18.18% |

## A.10. Computational Cost Analysis

Table 13 summarizes the computational costs of SWE-ABS compared to UTBoost across key dimensions.

*Table 13.* Cost-efficiency comparison between UTBoost and SWE-ABS. While SWE-ABS has higher cost per instance, its cost per strengthened instance is dramatically lower, resulting in substantially improved cost-efficiency.

| Metric | UTBoost | SWE-ABS |
|---|---|---|
| Avg. tokens per instance | 42,000 | 81,000 |
| Avg. wall-clock time (min) | 12.3 | 18.5 |
| Cost per instance ($) | 1.60 | 2.50 |
| Total cost for 500 inst. ($) | 800 | 1250 |
| Strengthened instances | 10 | 251 |
| Cost per strengthened inst. ($) | 80.00 | 4.98 |

**LLM Usage.** We evaluate SWE-ABS using two representative LLMs to assess framework effectiveness across different model capabilities: GPT-5 and GLM-4.7. Each model is applied independently to all pipeline components (test generation, decoupling, mutant synthesis, equivalence filtering), enabling controlled comparison. Using GPT-5, per-instance costs average $2.5, totaling approximately $1,250 for 500 instances. Using GLM-4.7, per-instance costs average $2.1, totaling approximately $110 for 50 instances.

**Runtime.** End-to-end processing averages 18.5 minutes per instance. Pipeline breakdown: program slicing (8%), test generation (42%), mutant synthesis (25%), equivalence filtering (5%), test execution (20%).

**Cost-Efficiency Analysis.** Although SWE-ABS incurs higher cost per instance than UTBoost ($2.50 vs. $1.60), it achieves substantially better cost-efficiency when effectiveness is measured by the cost per strengthened instance. Specifically,

UTBoost requires $80.00 to successfully strengthen one instance (10 strengthened instances at a total cost of $800), whereas SWE-ABS achieves this at a cost of $4.98 (251 strengthened instances at a total cost of $1,250), corresponding to a 16× reduction in cost per successful strengthening. This result highlights that SWE-ABS allocates computational resources more effectively by focusing on semantically critical and adversarial test scenarios, leading to significantly improved practical efficiency despite higher per-instance computational overhead.

### A.11. Qualitative Case Studies

A.11.1. TEST STRENGTHENING FAILURE

**Case Study: Overfitting failure (pytest-dev__pytest-7205).** The augmented tests encode the gold patch's implementation details as the mandatory contract. As a result, alternative correct fixes are rejected.

**Issue.** The `-bb` flag forces Python to treat implicit `bytes`-to-`str` conversions as fatal errors. Thus, pytest crashes when it implicitly converts byte parameters to strings during display. To resolve this issue, the rendering logic must be updated to explicitly handle binary data-for instance, by employing `saferepr`-to prevent runtime failures.

**Root cause.** `src/_pytest/setuponly.py` prints `fixturedef.cached_param` via `"[{}]".format(...)`; when it is `bytes`, implicit conversion triggers `BytesWarning`.

**Gold patch.** The fix in Listing 9 switches to `saferepr(...)` and chooses `maxsize=42`, which is sufficient to fix the bug but not mandated by the issue (default `saferepr` is 240).

*Listing 9.* Explicitly String Conversion

```
# src/_pytest/setuponly.py
+ from _pytest._io.saferepr import saferepr
  ...
-     tw.write("[{}]".format(fixturedef.cached_param))
+     tw.write("[{}]".format(saferepr(fixturedef.cached_param, maxsize=42)))
```

**Why the strengthened tests overfit.** The strengthened tests erroneously codify the visual output of the reference implementation as a functional requirement. As illustrated in Listing 10, the generated test asserts that the output length must not exceed 42 characters and strictly checks for specific formatting artifacts like hexadecimal representation.

Consequently, functional patches that safely handle the exception but utilize different visual formatting or length limits are incorrectly classified as invalid.

*Listing 10.* Overfitting Strengthened Test

```
def test_param_repr_maxsize_42_str(testdir):
    ...
    # Hard-coded assertion derived from the gold patch's maxsize=42
    assert len(content) <= 42

def test_setup_show_repr_exception_is_handled(testdir):
    ...
    # Asserts implementation detail (memory address format)
    assert "0x" in out
```

A.11.2. AGENT-GENERATED PATCHED FAILURE UNDER STRENGTHENED SUITES

Below we summarize six representative error categories.

**Category 1: Incomplete fixes** The patch addresses only part of the issue, missing some cases or scenarios.

**Instance ID.** sympy__sympy-12489

**Agent.** Bloop

**Issue.** SymPy's `Permutation` class employs an internal "array form" optimization where `Permutation._new_`

frequently calls `_af_new(...)` to generate objects. In the original implementation, `_af_new` was hardcoded to construct the `Permutation` class directly (via `Basic.__new__(Perm, ...)`). Consequently, subclassing `Permutation` failed to preserve the subclass type during instantiation or arithmetic operations, reverting objects to the parent class.

**Gold fix.** The correct solution (Listing 11) systematically replaces all internal hardcoded constructors with `cls` (the actual class reference) to support inheritance.

*Listing 11.* Correct implementation handling inheritance (Gold Fix)

```
# sympy/combinatorics/permutations.py
@@ -857,19 +860,19 @@ def __new__(cls, *args, **kwargs):
         if not args:
-            return _af_new(list(range(size or 0)))
+            return cls._af_new(list(range(size or 0)))
         if len(args) == 1:
             a = args[0]
-            if isinstance(a, Perm):
+            if isinstance(a, cls):
                 if size is None or size == a.size:
                     return a
-                return Perm(a.array_form, size=size)
+                return cls(a.array_form, size=size)
...
-    @staticmethod
-    def _af_new(perm):
+    @classmethod
+    def _af_new(cls, perm):
-        p = Basic.__new__(Perm, perm)
+        p = Basic.__new__(cls, perm)
```

The gold patch changes `_af_new` from a `@staticmethod` to a `@classmethod` and ensures that helper methods like `rmul_with_af` and `unrank_lex` utilize `cls` or `self` instead of the global `Perm` alias.

**Agent miss.** The agent (Listing 12) partially modified `__new__` but neglected to update the underlying object creation methods.

*Listing 12.* Incomplete agent patch missing factory methods

```
diff --git a/sympy/combinatorics/permutations.py b/sympy/combinatorics/permutations.py
@@ -857,19 +857,22 @@ def __new__(cls, *args, **kwargs):
         if not args:
-            return _af_new(list(range(size or 0)))
+            return cls._af_new(list(range(size or 0)))
```

**Why $T_{\mathrm{ori}}$ missed it.** Original tests cover common constructors. They do not stress subclassing through internal factories and operator returns.

**How $T_{\mathrm{aug}}$ closes the gap.** Add subclass-focused tests (e.g., `test_ops_return_subclass_instances`). Exercise the remaining factory path. Assert returned instance types. The agent patch fails.

**Category 2: Logical errors**   The fix logic is fundamentally wrong or uses incorrect algorithm/approach.

**Instance ID.** django__django-16527

**Agent.** Moatless Tools + Claude 4 Sonnet

**Issue.** In the Django admin interface, the "Save as new" functionality essentially duplicates the current object to create a new one. Semantically, this is an `add` operation (INSERT) rather than a `change` operation (UPDATE). Therefore, the button's visibility is currently controlled by has_change_permission, but it should be based on has_add_permission.

**Gold fix.** The fix replaces the permission check logic (Listing 13).

*Listing 13.* Correct permission logic replacement

```
diff --git a/django/contrib/admin/templatetags/admin_modify.py b/django/contrib/admin/
    ↪ templatetags/admin_modify.py
@@ -100,7 +100,7 @@ def submit_row(context):
            "show_save_as_new": not is_popup
-           and has_change_permission
+           and has_add_permission
            and change
            and save_as,
```

**Agent miss.** The agent fundamentally misunderstood the requirement, stacking the permissions instead of replacing them. The agent introduced a logical error by requiring *both* add and change permissions (Listing 14).

*Listing 14.* Flawed agent logic stacking permissions

```
--- a/django/contrib/admin/templatetags/admin_modify.py
+++ b/django/contrib/admin/templatetags/admin_modify.py
@@ -100,6 +100,7 @@
            "show_save_as_new": not is_popup
+           and has_add_permission
            and has_change_permission
```

**Why $T_{\mathbf{ori}}$ missed it.** Tests do not isolate the add+view-only role. They never assert visibility without change permission.

**How $T_{\mathbf{aug}}$ closes the gap.** Add a policy-targeted test (e.g., test_save_as_new_with_add_and_view_only). Expect the button to appear. The agent patch hides it. So it fails.

**Category 3: Boundary violations**    Edge cases, null checks, empty inputs not handled properly.

**Instance ID.** sphinx-doc__sphinx-8265

**Agent.** Sonar Foundation Agent + Claude 4.5 Sonnet

**Issue.** Sphinx utilizes an _UnparseVisitor to reconstruct Python code strings from AST nodes. A standard multidimensional subscript, such as A[1, 2], is represented in the AST as a Tuple slice. However, an empty tuple () represents a boundary condition: A[()] must be rendered as A[()], whereas rendering it as an empty string results in A[], which is syntactically invalid.

**Gold fix.** The correct fix (Listing 15) explicitly checks if the tuple elements are non-empty before stripping the parentheses.

*Listing 15.* Gold fix handling empty tuple boundary

```
diff --git a/sphinx/pycode/ast.py b/sphinx/pycode/ast.py
@@ -166,14 +166,28 @@ def visit_Subscript(self, node: ast.Subscript) -> str:
+       def is_simple_tuple(value: ast.AST) -> bool:
+           return (
+               isinstance(value, ast.Tuple) and
+               bool(value.elts) and ...
+           )
+       if is_simple_tuple(node.slice):
+           # Remove parentheses only for non-empty tuples
```

**Agent miss.** The agent (Listing 16) applied a broad transformation that flattened all tuple slices. The agent overlooked the empty set boundary. By unconditionally joining elements, Tuple(elts=[]) resulted in an empty string, producing A[] instead of A[()].

*Listing 16.* Agent patch overlooking boundary condition

```
+       if isinstance(node.slice, ast.Tuple):
+           elements = ", ".join(self.visit(e) for e in node.slice.elts)
+           return "%s[%s]" % (self.visit(node.value), elements)
```

**Why $T_{\text{ori}}$ missed it.** Tests cover typical tuple slices only. They omit the empty-tuple case.

**How $T_{\text{aug}}$ closes the gap.** Add an explicit empty-tuple test (e.g., `test_unparse_subscript_tuple_slices[A[()]]`). Assert the exact string. The agent patch fails.

**Category 4: Type mismatches**   Wrong type conversions, comparisons, or type mismatches.

**Instance ID.** django__django-10973

**Agent.** Tools + Claude 4 Opus (2025-05-22)

**Issue.** The task involved passing a database password to the `psql` utility via the `PGPASSWORD` environment variable. Environment variables in Python's `subprocess` module must be strings; passing integers or other types causes runtime errors or assertion failures.

**Gold fix.** The fix explicitly casts the password to a string (Listing 17).

*Listing 17.* Explicit type casting

```
+        subprocess_env = os.environ.copy()
+        if passwd:
+            subprocess_env['PGPASSWORD'] = str(passwd)
```

**Agent miss.** The agent passed the variable directly without type conversion (Listing 18).

*Listing 18.* Agent missing type conversion

```
+            if passwd:
+                env['PGPASSWORD'] = passwd
```

**Why $T_{\text{ori}}$ missed it.** Tests only use string passwords. No non-string input is exercised.

**How $T_{\text{aug}}$ closes the gap.** The test suite deliberately injected an integer password to verify robustness. The agent failed to sanitize the input for the environment variable interface, leading to a type mismatch (`123456 != '123456'`).

### A.12. Limitations and Threats to Validity

We identify three main limitations of the current approach.

**Test Overfitting Risk.**   Despite the test decoupling module (Section 3.1.2), SWE-ABS exhibits a 10.6% false-negative rate (53/500 instances) where augmented tests incorrectly reject valid alternative patches by encoding gold-patch-specific behaviors. Fully automated detection of such overfitting remains an open challenge. We provide a detailed case study in Appendix A.11.1.

**Analysis Scope.**   Our intraprocedural program slicing restricts analysis to statements within the same function or class as the patch (Section 3.1.3). This design choice trades precision for scalability but may miss cross-module dependencies (e.g., when a patch modifies a utility function called from multiple modules with different preconditions). Extending to interprocedural analysis would improve coverage identification at a computational cost that may be prohibitive for large-scale benchmarks.

**Gold Patch Dependency.**   SWE-ABS requires access to a gold patch for test generation, decoupling, and mutation synthesis. This limits applicability to scenarios with reference implementations and precludes use in settings where only issue descriptions are available, such as real-time bug triage or fully automated repair pipelines. Future work may explore weakening this requirement through property-based testing or specification mining techniques.

Beyond these limitations, we discuss broader threats to validity.

**Internal Validity.**   LLM-based components inherit risks of hallucination and prompt sensitivity.

**External Validity.**    Our evaluation focuses on four primary languages (Python, JavaScript, Go, TypeScript) covered by SWE-Bench and SWE-Bench Pro, specifically in strong copyleft licenses repositories. As a result, the observed effectiveness may not generalize to other programming languages outside this subset, proprietary codebases, functional programming paradigms (e.g., Haskell), or low-level systems languages (e.g., C). Nevertheless, the consistent improvements observed across Verified and Pro indicate that our approach addresses fundamental test inadequacy rather than being limited to benchmark-specific artifacts.

**Construct Validity.**    Our strengthening metric operationalizes "incorrect patch" as one that passes $T_{\text{ori}}$ but fails $T_{\text{aug}}$, which assumes the augmented tests do not overfit to gold-patch-specific behaviors. We quantify this risk through false-negative analysis (Section 4.4.1), finding a 10.6% overfitting rate. Additionally, our LLM-based equivalence annotation (Section 3.2.2) may misclassify subtle semantic differences; we mitigate this through majority voting across three independent LLMs.

