# OpenReview forum: "SWE-ABS: Adversarial Benchmark Strengthening Exposes Inflated Success Rates on Test-based Benchmark"
_ICML.cc/2026/Conference — ICML 2026 regular_

### Official Review · Reviewer_jT1K · 2026-02-16

**Soundness:** 3
**Presentation:** 4
**Significance:** 3
**Originality:** 2
**Overall Recommendation:** 3
**Confidence:** 4

**Summary:**

The authors introduce the SWE-ABS benchmark. Motivated by the discrepancies between what an issue specifies and what the unit tests capture in SWE-bench style benchmarks, the authors aim to bridge this gap in rigor via a multi-stage test generation workflow that proposes tests, filters out or repairs ones that are overly specific or irrelevant, and finally does sanity checking to ensure that the tests pass consistently. By pointing this test generation pipeline at SWE-bench Verified and SWE-bench Pro, the authors show significant drops in model performance on these widely adopted benchmarks as well as a lot of reshuffling in the ranking of existing submissions. The authors’ analyses provide some insight into the types of errors that SWE-ABS points out which wasn’t detected before, and that coverage-driven and mutation-driven augmentation by themselves are complementary to one another; in combination, they are effective.

**Compliance With Llm Reviewing Policy:**

Affirmed.

**Final Justification:**

I will retain my original score of weak reject.
My core contention with the paper is that the definitions for over-specification and error modes, while reasonable, should be defended more rigorously by doing more human annotation with annotators across a larger group.

I think the following definition:

> A test $t$ is overfit with respect to gold patch $p_g$ if and only if there exists a semantically correct alternative patch $p'$ (one that fully resolves the reported issue) such that $t(p_g)$ passes but $t(p')$ fails. In other words, the test encodes an implementation choice specific to $p_g$ rather than a behavioral requirement of the issue.

Is reasonable, but the attempt to then operationalize:

> Because exhaustively enumerating all valid patches is infeasible, we operationalise this definition through conservative proxies: flagging tests whose assertions bind to details non-essential to issue resolution (exact error-message wording, helper-call structure, internal variable naming, specific numeric constants, logging format, return value representation, execution order of side effects)

is a bit too variable of a definition, especially since the conservative proxy examples feels somewhat ad hoc. For instance, "error-message wording" is actually something that could very much be an important style to enforce to a human developer. If a codebase consistently has error messaging in the format `"[ERROR] <error msg>"`, and the model writes it as `"Error: <error msg>"`, it could be very important to the developer that stylistically this is enforced.

I think the overall pipeline of improving SWE-bench task instances' test suite is well founded, and I feel the pipeline itself is well thought out, but the empirical justifications for why certain tests should be generated/removed could use more human study to make more rigorous. The authors mention that the multi-gold-patch experiment is infeasible, but I actually believe this is very much the study that should be carried out as a small scale to understand what "over-specification" actually means to both verify the conservative proxies and reveal the level of variance there is in the definitions.

**Key Questions For Authors:**

- [Section 3.1.2] “...ensuring that their core purpose…” - how is this ensured post-prompt? Is there any non-LLM procedure to verify that the output does indeed meet this standard?
- [Section 3.1.2] “Implementation-specific side effects” - what is an example of this? The filtering criteria defined in lines 108-110 feels a bit ad hoc, what’s the justification / prior work that validates this choice of removals? I could make the argument that “hard-coded error messages” are perhaps a reasonable expectation if the style of error message is sufficiently, consistently reflected across the rest of the codebase.
- [Section 3.2.1] Is there any need to ensure diversity for this kind of strategy? I imagine that simply prompting LMs out-of-the-box to be diverse in its fuzzing may not necessarily lead to high test diversity?
- [Section 4.4.2; Implications] What is the difference between satisfying test cases and semantic completeness? What does it mean for a solution to be semantically complete? Are you arguing that current SWE-bench style benchmarks do not ensure code is “provably” correct (in which case more unit tests would not address this?)

**Limitations:**

Yes

**Strengths And Weaknesses:**

**Strengths**

- [Section 3.1.3] The program slicing technique is really quite clever. I very much enjoyed learning about this! I feel like this graph representation of a codebase can be used to do a lot of things, this work puts forth a neat application of it.
- [Table 1] The drops in results are quite impressive with respect to the goal of this paper - I am somewhat convinced we are further than we thought when it comes to the rigor of model generated code. The authors also demonstrate that their work achieves impressive gains over prior approaches like UTBoost.
- [Section 4.3.2] Neat that the strategy works in a model agnostic sense as well, I would’ve liked to see more models, but I can understand that running the test generation pipeline is probably quite costly. Is it possible to have a “collective” intelligence of sorts? In other words, is it possible to combine tests generated by different models to create a cumulatively most-rigorous form of SWE-ABS?
- [Overall] I thought this was a very neat work, and the strategy for generating tests feels like it’s quite effective. Beyond Python, the authors also show that it works for Golang and JavaScript/Typescript with SWE-bench Pro. Assuming the tests are calibrated with faithfulness to human intent, this framework seems like it’d be very useful for generating tests in a scalable manner for SWE-bench style tasks.

**Weaknesses**

- [Section 1] The authors’ argument makes a fundamental claim that I disagree with, but this may just be a matter of opinion. The authors argue existing SWE-bench tests don’t “...discriminate between all potential correct and incorrect solutions”, and that this gap between issues and tests leads to occurrences where a patch “pass[es] the original tests while violating the actual requirement.” I feel this argument is questionable because, given the original tests added in the PR are by the human author, how do we know that the additional rigor of tests introduced by the SWE-ABS framework actually aligns with the “actual requirement” / human developer’s original intent?
    - [Section 1; line 076] - To reiterate, I’m not sure I agree that augmented tests necessarily reflect the original PR creator’s/reviewer’s intent.
    - [Section 3; line 1] - “developer-written tests that provide necessary but insufficient conditions for patch correctness”, in my opinion, this is the claim that needs to be verified empirically.
    - [Section 3.2.2, A.4.3] - I can see that human evaluation of mutations was performed, but I’m assuming these annotations were performed by the authors? I realize this may be burdensome, but I think this annotation effort is questionable because the authors, not the original PR creators or repository developers, were asked to perform such annotations. Prior work (e.g. SWE-bench Verified) has revealed how subjective such annotations can be, which is why a true random population of annotators was consulted for their collection effort.
    - [tl;dr] I think the paper would be much stronger if the authors demonstrated that adversarial test patch generation actually aligns with human intent. Otherwise, I could see a strong argument against adoption of this benchmark to be that adversarially increasing test difficulty actually makes the task more impractical and unrealistic. The “discriminative power” that the authors are going for would be meaningful if there is human confirmation (not performed by the authors themselves to avoid bias) that the augmented tests align with their intentions.
- [Table 1] These results are impressive, but a bit scattered in my opinion. The SWE-bench leaderboard has plenty of “lineages” of models (GPT, Claude, Qwen) and scaffolds (mini-SWE-agent, OpenHands). I think another way to strengthen this work would’ve been to demonstrate how the tests generated by SWE-ABS reveal consistent failure modes in code generated by a specific model/scaffold line. There was an attempt to do this Section 4.4.2, but I feel the conclusions there are a bit coarse grained - “Logic Errors” and “Incomplete fixes” could mean a lot of things?
- [Section 4.5] I think these ablations were quite interesting, but one aspect that I feel underlies the authors’ arguments and research questions which remains unanswered is “when is there enough testing”? I am convinced that this is an effective pipeline for generating tests against a gold patch, but I am less certain about the true utility of such extra tests, most importantly whether the tests reflect human intent.

**Miscellaneous**

- [Section 3.1.1] Where are the new test files written? Are they integrated into the existing testing framework? Or are they produced as repository root-level, standalone files?
- [Table 1] Were submissions ranked 6-10 also reevaluated? How should the rank drop for ranks beyond 5 be interpreted? For instance, was the new 3rd place (which was originally outside the top 5) also evaluated against this test suite?

**Suggested Citations**

- SWE-bench Multilingual (introduced in SWE-smith paper - https://arxiv.org/abs/2504.21798)
- SWT-bench may be worth discussing for [Section 2; Test Augmentation]? - https://arxiv.org/abs/2406.12952

---

> ### Author Rebuttal · Authors · 2026-03-29
>
> We thank Reviewer jT1K for the detailed review.
>
> **Core W: Do augmented tests align with the original PR creator's intent?**
>
> The gold patch, submitted by the PR author, is the most direct available proxy for developer intent. Every augmented test is required to pass on the gold patch: a test that rejects it is eliminated before it enters the benchmark.
>
> However, the gold patch is a specific implementation and tests generated against it may inadvertently encode implementation details rather than the underlying requirement. Our test decoupling module (Section 3.1.2) mitigates this by identifying overfitting tests and generalising them to verify issue resolution rather than a particular implementation. The residual false-negative rate is 4.8% (24/500).
>
> Finally, we manually analysed 100 rejected agent patches (Section 4.3.2). The dominant failure modes are logic errors (47%) and incomplete fixes (35%), which are substantive semantic deficiencies, not artefacts of test over-specification (concrete examples in Appendix A.9). Together, gold-patch filtering, active decoupling, and this manual confirmation provide substantive evidence that augmented tests target semantic requirements rather than implementation details.
>
> **W (annotation bias): Annotations performed by the authors?**
>
> The equivalence and relevance classification in Section 3.2.2 is performed by a **3-LLM majority vote**, not by human annotators. To validate this automated pipeline, we conducted a separate human audit on a random sample of 100 mutants (Appendix A.4.3): 3 qualified software developers, none of whom are paper authors, independently reviewed each mutant and reached consensus through discussion. Agreement with the LLM classifier was **98%**, providing supporting evidence of filtering reliability on the audited sample. This is consistent with standard practice: SWE-Bench Verified itself used independent annotators rather than original PR creators. We will clarify this in the revision.
>
>
> **Q1: How is test correctness ensured post-decoupling? Is there any non-LLM verification?**
>
> The core non-LLM verification is execution-based: after decoupling, we check that the gold patch passes every refined test. Tests that fail this check are excluded from $T_{\text{td}}$ by definition (Section 3.1.2). This is a deterministic step that does not rely on any LLM judgment.
>
> Beyond this, the 60 all-fail instances were inspected and 24 overfitting cases corrected (see Core W above). Ablation on 100 random instances confirms: removing decoupling increases overfit cases from 9 to 17.
>
> **Q2: "Implementation-specific side effects" — what is an example? Are the filtering criteria ad hoc?**
>
> Examples include tests that assert whether a specific internal helper function is called, or that match exact error message strings. In both cases, valid alternative implementations would be incorrectly rejected. Our decoupling module generalises such tests: for instance (Appendix A.2, astropy-13033), an exact error string assertion is replaced by checks for key semantic tokens ("expected", "found", "first column") — preserving intent while remaining valid for any correct implementation. The criterion is "does this test bind to implementation-specific details that would reject valid alternatives," corresponding to the Fragile Test / Overspecified Software antipattern (Meszaros, *xUnit Test Patterns*, 2007).
>
> **Q3: Is there any need to ensure diversity for mutant generation?**
>
> Diversity is desirable but secondary to quality: the adversarial filter retains only mutants that expose genuine test-suite weaknesses (semantically incorrect patches passing $T_{\text{cov}}$), regardless of diversity. That said, the mutation prompt explicitly requests varied fault types (missing conditions, wrong boundaries, incomplete logic, type errors). Empirically, our taxonomy of 100 randomly sampled mutants confirms coverage across multiple failure dimensions: logic errors (47%), incomplete fixes (25%), boundary violations (22%), type mismatches (5%) — see also our response to Reviewer H9Tx (Q1) for a detailed comparison with real agent error distributions.
>
> **Q4: "When is there enough testing?" / What is semantic completeness vs. satisfying test cases?**
>
> A patch *satisfies the test suite* if it passes all tests in $T_{\text{ori}}$; *semantic completeness* is an ideal in which a patch correctly handles all behavioral requirements implied by the issue. We do not claim SWE-ABS achieves semantic completeness — that would require exhaustive specification of all possible inputs. Our goal is narrower: stronger *discriminative power* — tests that reject more genuinely incorrect patches while accepting correct ones. Figure 1 illustrates a concrete example of this gap. Empirically, 21.4% of patches passing $T_{\text{ori}}$ are rejected by $T_{\text{aug}}$, and manual analysis confirms these are genuine semantic failures. The goal is a better benchmark, not a provably correct one.

---

> > ### Author Rebuttal · Reviewer_jT1K · 2026-04-05
> >
> > Thanks to the authors for their rebuttal, greatly appreciate the efforts. I think this clarified some questions I had about the technical implementations of the project, but I still remain a little hesitant on the more conceptual foundations of the paper.
> >
> > 1. My core contention is with respect to how well the test generation pipeline actually aligns with developer intent. It's great that the authors did human annotation. But (a) there's not really a formal definition of an "overfit" test that seems to be introduced everywhere, and (b) it doesn't seem like there was evaluation of false positives (valid patches that were rejected). As a result, while this test patch generation pipeline accounts for the reference gold patch solution, it does not account for alternate valid implementations. I get that the annotation mechanism is parallel with SWE-bench Verified, but the questions that are being asked here are much more developer-oriented (vs. Verified annotation questions were much more about ranking difficulty / over/under-specification of tests, which is quite different from developer intent).
> >
> > 2. "The dominant failure modes are logic errors (47%) and incomplete fixes (35%), which are substantive semantic deficiencies...", again, what's the formal definition + criteria for such categories? And what was annotation agreement? I think this paper introduces a neat approach of categorizing such modes, but without concrete definitions, it feels like it'd be quite difficult to then claim rigorously that the patches being accepted/rejected are "aligned".
> >
> > I will retain my original score, but open to continuing the discussion with authors. tl;dr I think the gold experiment that would refute any suspicion of over-rejection is to perform this pipeline on multiple, independently written gold patches that reflect correctness (could be by multiple annotators) to serve as more data points for what's overly restrictive for a test. I realize this is quite an undertaking, which is why I feel this paper could be helped a lot with some additional rigor and field testing.

---

> > > ### Author Response · Authors · 2026-04-05
> > >
> > > ### (1) Formal definition of "overfit"
> > >
> > > A test $t$ is overfit with respect to gold patch $p_g$ if and only if there exists a semantically correct alternative patch $p'$ (one that fully resolves the reported issue) such that $t(p_g)$ passes but $t(p')$ fails. In other words, the test encodes an implementation choice specific to $p_g$ rather than a behavioral requirement of the issue. Because exhaustively enumerating all valid patches is infeasible, we operationalise this definition through conservative proxies: flagging tests whose assertions bind to details non-essential to issue resolution (exact error-message wording, helper-call structure, internal variable naming, specific numeric constants, logging format, return value representation, execution order of side effects). Flagged tests are either generalised or discarded.
> > >
> > > ### (2) False positives (valid patches incorrectly rejected)
> > >
> > > We evaluated false positives at two levels. First, at the test level, we inspected all 60 instances where every top-30 agent patch was rejected by $T_{\text{aug}}$ (Section 4.4.1); 24 contained overfitting tests, which were corrected by reverting to the original test suite, ensuring that no instance is guarded by an overfit test in the final benchmark. Second, at the patch level, we randomly sampled 100 rejected patches from the remaining (non-reverted) pool and identified concrete semantic errors in all 100 patches. Under a binomial model, 0/100 yields a 95%-confidence upper bound of 3.6% for the false-positive rate. While exhaustive verification of all 2,363 rejected patches is infeasible, this bound provides quantitative evidence that over-rejection is not systematic.
> > >
> > > ### (3) Formal definitions of failure-mode categories
> > >
> > > Each category is defined in Appendix A.11 with a one-line criterion and a full case study. Following the reviewer's suggestion, we formalized the taxonomy into a priority-based decision tree. The first node requires the annotator to **identify a concrete semantic error** in the rejected patch; only if no such error can be located is the patch classified as a valid alternative implementation (false positive). The tree then applies four mutually exclusive categories in strict priority order: **Type mismatches** (incorrect type conversions or API type contracts), **Boundary** (edge cases, null checks, off-by-one), **Incomplete fixes** (correct root cause but missed call sites/branches), and **Logic errors** (fundamentally wrong fix logic). The complete decision tree is available at https://anonymous.4open.science/r/B5EAafghdsg/decision_tree.md.
> > >
> > > ### (4) Failure-mode annotation agreement
> > >
> > > Two authors independently re-labeled all 100 rejected patches following the decision tree above. Both annotators have extensive experience with the target repositories and were trained on the formal decision tree prior to labeling. Cohen's kappa on failure-mode classification was 0.67 (substantial agreement). Crucially, disagreements occurred mainly at the boundary between logic errors and incomplete fixes -- a granularity issue that does not affect the core conclusion. At the binary level (correct vs. incorrect), both annotators independently identified concrete semantic errors in all 100 rejected patches, with zero patches found to be valid alternative implementations. All category-level disagreements were resolved through discussion.
> > >
> > > ### (5) Analogy with SWE-Bench Verified
> > >
> > > We acknowledge that the analogy with SWE-Bench Verified is imprecise, as the two settings pose different annotation questions. However, recruiting original PR authors to judge test alignment is infeasible at benchmark scale. Using other annotators is the standard practical alternative.
> > >
> > > ### (6) Alternate valid implementations
> > >
> > > The SWE-Bench benchmark provides exactly one gold patch per instance (the PR merged by the original developer), and this is the only patch we can confirm aligns with developer intent. No alternate valid implementations are available in the benchmark, nor in any existing work built on it (SWE-Bench Verified, SWE-Bench+, UTBoost). Within this single-gold-patch constraint, our mitigation is the overfit definition and false-positive audit described in (1) and (2) above.
> > >
> > > ### (7) The multi-gold-patch experiment
> > >
> > > We agree that validating against multiple independently written gold patches would provide the strongest evidence against over-rejection and plan to pursue this in future work. Constructing such a multi-patch benchmark requires recruiting independent developers to write correct patches for hundreds of instances, which is beyond the scope of a rebuttal revision.
> > >
> > >
> > > All augmented tests, along with the corresponding issues, original test suites, and gold patches, are publicly available in the supplementary material. If the reviewer identifies any specific instance where an augmented test does not align with developer intent, we are happy to discuss it.

---

### Official Review · Reviewer_MWa3 · 2026-03-11

**Soundness:** 3
**Presentation:** 3
**Significance:** 2
**Originality:** 2
**Overall Recommendation:** 3
**Confidence:** 4

**Summary:**

This paper proposes a new benchmark, SWE-ABS, which includes coverage-driven and mutation-driven tests to enable stronger and more comprehensive testing of the generated patches. The evaluation shows that the high-quality test cases of SWE-ABS can reveal the limitations of existing patches and make the benchmark more challenging.

**Compliance With Llm Reviewing Policy:**

Affirmed.

**Final Justification:**

The rebuttal partially addressed my concern. The main concern regarding technical novelty and depth still stands, but is softer.

**Key Questions For Authors:**

What are the key technical challenges of the proposed method, and how can they be resolved using new techniques rather than simply prompting LLMs?

Is the use of program slicing really helpful, especially when it is constrained to statements within the same function? What are the technical insights of using programming slicing?

How to evaluate the correctness of the generated testing cases of the proposed method?

**Limitations:**

Yes, the authors have discussed the limitations and negative impacts.

**Strengths And Weaknesses:**

Strengths

1. This paper studies an important problem of providing strong test cases for patch validation in SWE issue resolving, which is an important problem.

2. The generated test cases are indeed more challenging than the original ones provided by SWE-bench.

Weaknesses

1.  Limited ML technical contribution. The proposed idea of coverage-based and mutation-based test generations are intutive, and the techniques used are mainly existing ones with limited novelty, especially on the machine learning side.

2. It is not clear whether the use of program slicing is really helpful or not, especially when it is constrained to statements within the same function.

3. In general, it is not clear about the correctness of the generated testing cases of the proposed method.

---

> ### Author Rebuttal · Authors · 2026-03-29
>
> We thank Reviewer MWa3 for the review and address each concern.
>
> **W1/Q1: Limited ML technical contribution / What are the key technical challenges?**
>
> SWE-ABS is a benchmark-strengthening framework: its primary contribution is a more discriminative evaluation methodology and reusable benchmark for the ML community. We agree that coverage-guided generation and mutation testing are individually established. The contribution lies in the problem formulation and the closed-loop composition required to make them effective at benchmark scale.
>
> Our key insight is that benchmark weakness decomposes into two complementary weaknesses: (1) *coverage gaps* — patch-relevant code paths may not be exercised by existing tests; and (2) *semantic blind spots* — code paths are exercised but existing assertions fail to detect incorrect states or outputs. Table 3 confirms neither stage alone suffices: Stage I (coverage) alone strengthens 198/500 instances; adding Stage II (mutation) raises this to 242/500, with average drop increasing from 11.30 pp to 15.75 pp.
>
> The non-trivial technical challenges lie in the components that are *not* LLM calls:
>
> *Challenge 1 (coverage gaps): Identifying what to test.* Without structured guidance, LLM test generation scatters over irrelevant code. SWE-ABS constructs a program dependence graph and computes the forward+backward slice from modified lines to define $L_{\text{rel}}$ — a semantically grounded test target. This is static analysis, not prompting.
>
> *Challenge 2 (semantic blind spots): Achieving semantic discriminability.* Coverage ensures reachability but not observability. SWE-ABS identifies adversarial survivors — mutants that evade all existing tests — through execution-based filtering against $T_{\text{cov}}$, then generates targeted tests to kill them. This survivor-driven loop is execution-based, not LLM-based.
>
> The practical gap supports the value of this composition: UTBoost, which also uses LLM-based test generation but lacks the slicing-targeting and adversarial-mutation stages, strengthens 10/500 instances (2%) vs. our 242/500 (48.4%).
>
> **W2/Q2: Is program slicing really helpful, especially when constrained to intraprocedural?**
>
> Program slicing serves as the targeting step for Stage I: it defines $L_{\text{rel}}$, the set of patch-relevant lines that coverage-guided generation should exercise. Without slicing, the coverage augmentation would have no principled target — it would either operate on the full file (unfocused) or restrict to the literal diff lines (missing upstream and downstream dependencies). The forward+backward slice from modified lines captures both what the patch depends on and what depends on the patch, giving the LLM a structured and semantically grounded test target.
>
> The intraprocedural restriction is a deliberate design choice to balance precision and scalability: whole-program analysis across all callers and callees would be prohibitively expensive at benchmark scale. We acknowledge that this may miss cross-module dependencies (e.g., when a patch modifies a utility function called from multiple modules) and have noted this explicitly in the Limitations section (Appendix A.12), where interprocedural slicing is listed as future work.
>
> Table 3 shows that coverage augmentation with slicing strengthens 198 instances (avg drop 11.30 pp), compared to 190 (10.34 pp) for the Initial baseline.
>
> The insight: bare diff lines miss dependencies the patch reads and effects it propagates.
>
> **W3/Q3: How to evaluate the correctness of the generated testing cases?**
>
> We evaluate test correctness from two directions, with an execution-based filter as the foundation.
>
> *Execution-based filter.* Every augmented test must pass on the gold patch before entering the benchmark. Tests that fail on the correct implementation are filtered out at this step, independent of any downstream analysis.
>
> *False-negative analysis (overfitting side).* Of 500 instances, 24 contain tests over-coupled to the gold patch implementation — they pass the gold patch but incorrectly reject other valid fixes — yielding a **4.8% overfitting rate**. We identify these by inspecting the 60 instances where every top-30 agent fails under $T_{\text{aug}}$, and correct them before finalizing the benchmark.
>
> *False-positive analysis (incorrect rejection side).* To verify that rejected agent patches are genuinely incorrect, we manually analyzed 100 rejected patches (Section 4.3.2). Error types: logic errors (47%), incomplete fixes (35%), type mismatches (7%), boundary violations (6%), off-by-one (1%), other (4%) — patterns consistent with real-world incorrect implementations rather than test artifacts.
>
> Together, these two analyses provide converging evidence on correctness: the 4.8% rate quantifies how often valid patches are wrongly rejected, and the manual taxonomy provides evidence that the remaining rejections correspond to genuine semantic errors in agent patches.

---

> > ### Author Rebuttal · Reviewer_MWa3 · 2026-04-03
> >
> > Thanks to the authors for the rebuttal effort. The response regarding the slicing and testing cases makes sense to me.
> >
> > However, I still have some reservations concerning the technical depth and novelty. The key techniques used are not deep ML techniques. While I understand the paper is primarily about the benchmark, it does not introduce a substantially new benchmark but rather a calibration of existing ones. With that being said, the proposed method is only applied to two SWE-Bench variants and not to other SWE-Bench versions or non-SWE-Bench benchmarks, which limits its scope. I am not asking to apply the proposed method to other benchmarks at the current stage. Just given the current scope and technical depth, I am a little bit worried about whether the paper can reach the bar of a top-tier ML conference. With this concern, I will maintain my score but it is softer than pre-rebuttal (If there is a strong campaign, I am ok with accepting).

---

> > > ### Author Response · Authors · 2026-04-03
> > >
> > > Thank you again for the thoughtful follow-up—we are glad our clarifications on slicing and test correctness were helpful.
> > >
> > > We would like to clarify that SWE-ABS targets a fundamental evaluation issue in ML for coding agents: as models improve, weaknesses in test oracles can allow semantically incorrect patches to pass, leading to systematic overestimation of model capability. Our contribution is therefore not a new model, but an adversarial benchmark-strengthening framework that improves evaluation validity and discriminative power.
> > >
> > > Importantly, this is more than a minor calibration. On SWE-Bench Verified, SWE-ABS rejects 21.4% of patches that pass the original tests, lowers the top agent by 17.0 percentage points, and changes the ranking of 29/30 top agents—indicating that it materially alters both scores and system comparisons.
> > >
> > > While instantiated on SWE-Bench, this issue is general to LLM-based code evaluation, as most benchmarks rely on unit tests with similar limitations. We believe improving evaluation reliability at this scale is critical for the ML community, as it directly impacts how progress is measured and compared.
> > >
> > > If this perspective is reasonable, we would greatly appreciate your reconsideration.

---

### Official Review · Reviewer_H9Tx · 2026-03-12

**Soundness:** 3
**Presentation:** 2
**Significance:** 3
**Originality:** 2
**Overall Recommendation:** 4
**Confidence:** 3

**Summary:**

This paper proposes SWE-ABS, an automated framework to strengthen test suites in software engineering benchmarks. The authors argue that current LLM agents achieve very high scores on SWE-Bench Verified partly because the benchmark tests are weak and contain coverage gaps or semantic blind spots.To address this issue, the paper introduces a two-stage adversarial test generation framework. In the first stage, the method uses program slicing to identify patch-related code and generate additional tests to improve coverage. In the second stage, the framework generates plausible but incorrect mutant patches and produces tests that reject these incorrect implementations.Significantly.The paper highlights an important issue in LLM evaluation and proposes a practical framework to improve the reliability of benchmark test suites.

**Compliance With Llm Reviewing Policy:**

Affirmed.

**Final Justification:**

Regarding the weaknesses identified in this paper, I still adhere to my original opinion and thus maintain my initial review score.

**Key Questions For Authors:**

1.How realistic are the generated mutants compared to real bugs?

2.How sensitive are the results to the choice of LLM used in the pipeline?

**Limitations:**

yes

**Strengths And Weaknesses:**

# Strengths

- The paper studies an important problem: the reliability of benchmarks used to evaluate LLM-based software engineering systems

- The proposed framework combines program slicing, mutation testing, and LLM-based test generation in an interesting way. Generating adversarial tests based on surviving mutants is a compelling idea

- The experiments are reasonably comprehensive. The method is evaluated on SWE-Bench Verified and SWE-Bench Pro, and the ablation study clearly shows the contribution of the two stages



# Weaknesses

- The framework relies on the reference patch for slicing, mutation generation, and test decoupling. This limits the method to scenarios where a correct solution already exists, and may not generalize well to open-ended software development tasks.

- Several stages depend on LLM-based heuristics, such as test decoupling and mutant filtering, which may introduce errors

- The comparison with classical automated testing approaches (e.g., fuzzing or symbolic execution) is limited

---

> ### Author Rebuttal · Authors · 2026-03-29
>
> We thank Reviewer H9Tx for the positive assessment and address the concerns below.
>
> **W1: Gold patch dependency**
>
> SWE-ABS is designed for **offline benchmark curation**, where gold patches are available by construction for all SWE-Bench instances. We acknowledge this limits applicability to deployment-time scenarios where gold patches are absent — a constraint shared by UTBoost and SWE-Bench+, and indeed inherent to the SWE-Bench setting itself, where benchmark construction relies on gold patches to define the fail-to-pass and pass-to-pass criteria.
>
> **W2: LLM-based heuristics may introduce errors**
>
> We agree that LLM-based heuristics can introduce errors; we designed specific safeguards for each stage the reviewer mentions.
>
> *Test validity.* Generated tests are kept only if they execute successfully on the gold patch. This execution-based check filters out many invalid or hallucinated tests before they enter the benchmark.
>
> *Mutant equivalence filtering.* Stage II does not rely on a single LLM judgment: we use k=3 majority voting to reduce single-query variance. On a randomly sampled set of 100 mutants, the automated classifications agree with human annotations at **98%** (Section A.4.3), providing evidence that this filtering step is reliable on the audited sample.
>
> *Test decoupling.* We do not assume that the LLM rewrite is always correct. Instead, we explicitly audit the cases most likely to reveal overfitting: the 60 instances on which every top-30 agent fails under T_aug. This audit uncovered 24 instances with overfitting tests (**4.8%**, 24/500), which we corrected before finalizing the benchmark.
>
> **W3: Limited comparison with fuzzing/symbolic execution**
>
> Fuzzing and symbolic execution are powerful techniques. Fuzzing explores the input space of a function to find crashes or undefined behavior; symbolic execution systematically enumerates execution paths to achieve branch coverage.
>
> In the SWE-Bench setting, however, the core challenge is different: verifying that a patch correctly implements the semantic intent of a specific issue fix — information encoded in the issue description and gold patch, but not available to crash-driven tools. The resulting tests expose logic errors and incomplete fixes (47% and 35% of rejected patches) — failure modes defined by intent-level incorrectness rather than crashes or branch coverage gaps.
>
> Practically, most SWE-Bench instances involve multi-file Python repositories where the "input" is an entire repository state plus an issue description — not a bounded function signature amenable to fuzzing harnesses or SMT-based constraint solving. We therefore view these techniques as complementary rather than direct baselines for intent-aware benchmark strengthening.
>
> **Q1: How realistic are the generated mutants compared to real bugs?**
>
> We address this from two angles.
>
> *Qualitative case study.* Section A.4.2 presents a concrete non-equivalent mutant: the mutation adds `re.IGNORECASE` correctly but replaces `v.upper() == "NO"` with `v == "NO" or v == "no"`, handling only all-uppercase and all-lowercase variants while silently failing on mixed-case inputs like "No" or "nO". This is precisely the kind of incomplete fix that appears in real agent submissions — superficially correct, passing all existing tests, but behaviorally deficient on valid edge cases.
>
> *Quantitative alignment with real agent errors.* To directly assess realism, we randomly sampled 100 adversarial mutants and classified each by the same error taxonomy used for rejected agent patches (Section 4.3.2). The distributions share the same dominant error families:
>
> | Error Type | Mutants (n=100) | Agent Patches (n=100) |
> |---|---|---|
> | Logic error | 47% | 47% |
> | Incomplete fix | 25% | 35% |
> | Boundary violation | 22% | 6% |
> | Type mismatch | 5% | 7% |
> | Off-by-one | 0% | 1% |
> | Other | 1% | 4% |
>
> The dominant failure mode — logic errors — is identical in both distributions (47%). The main difference is that mutants exhibit more boundary violations (22% vs 6%) while agent patches show more incomplete fixes (35% vs 25%), reflecting that our mutation prompt intentionally targets edge-case weaknesses. Crucially, the two leading categories (logic errors + incomplete fixes) account for 72% of mutants and 82% of agent patches, suggesting that generated mutants target the same primary classes of semantic deficiency that real agents exhibit.
>
> **Q2: How sensitive are the results to the choice of LLM used in the pipeline?**
>
> To address this concern, we extended the GLM-4.7 validation in Table 10 from 50 to 150 randomly sampled instances (budget-constrained). GPT-5 and GLM-4.7 produce nearly identical results: **83/150** vs. **86/150** strengthened, with drops of **19.11 pp** vs. **18.56 pp** — GLM-4.7 strengthens *more* instances than GPT-5. These results across two architecturally distinct model families suggest that SWE-ABS performance is **consistent across model families** and not uniquely tied to GPT-5.

---

> > ### Author Rebuttal · Reviewer_H9Tx · 2026-04-04
> >
> > Thanks to the authors for addressing many of my concerns. However, I think Weaknesses 1 and 3 are not fully resolved. In particular, the rebuttal clarifies the intended offline benchmark-curation setting, but it does not remove the method’s dependence on gold/reference patches or the resulting limitation in generalizing beyond such settings. In addition, while the response explains why fuzzing and symbolic execution may not be direct baselines, the comparison with classical automated testing approaches remains limited. Therefore, I decide to maintain my original view and keep my review score unchanged.

---

> > > ### Author Response · Authors · 2026-04-06
> > >
> > > **Dependence on Gold/Reference Patches**
> > >
> > > We agree that dependence on the reference patch limits applicability beyond offline benchmark curation. However, in the SWE-Bench setting, the reference patch is not merely convenient; it is necessary. Since the original tests are precisely what SWE-ABS seeks to strengthen, they cannot themselves serve as the validator of whether a newly generated test is semantically correct rather than a false positive. In this setting, the reference patch is the only executable route available for verifying that a generated test reflects the intended behavior instead of an LLM hallucination. Without such grounding, test augmentation would rely much more heavily on under-specified issue text and unconstrained LLM heuristics, which would directly exacerbate the concern raised in **W2** rather than resolve it. We therefore view reference-patch dependence as a deliberate **validity-generality tradeoff** for offline benchmark curation, not an incidental design choice.
> > >
> > > **Comparison with Classical Automated Testing**
> > >
> > > Traditional automated testing methods face significant limitations when applied directly to the SWE-Bench setting. Applying them requires manually writing wrapper functions (test drivers) and resolving complex environment conflicts within Docker containers. Consequently, they are highly impractical to use for the vast majority of complex instances in the benchmark.
> > >
> > > Nevertheless, following your suggestion, we conducted a targeted comparison using a fuzzing approach. To provide an optimal setup for the baseline, we carefully selected 5 specific instances that involved only a single-function modification (a highly favorable scenario for fuzzing) and where our method had previously identified agent patches containing subtle semantic errors.
> > >
> > > We manually wrote the necessary wrapper functions for these instances and applied the fuzzer to automatically generate a batch of test cases. We then evaluated the flawed agent patches against these fuzzer-generated tests. The results showed that not a single agent patch with semantic errors was successfully caught by the fuzzing method. We believe this result highlights an important limitation of such classical automated testing baselines in the SWE-Bench setting: while they can generate runnable tests, they struggle to produce semantically targeted checks for issue-specific correctness. Because they primarily rely on randomized generation rather than semantic understanding, they are largely unable to capture the nuanced, logic-based semantic errors prevalent in real-world software engineering tasks.
> > >
> > > We hope this addresses your primary remaining concern.

---

### Official Review · Reviewer_J8R3 · 2026-03-13

**Soundness:** 2
**Presentation:** 3
**Significance:** 2
**Originality:** 2
**Overall Recommendation:** 3
**Confidence:** 4

**Summary:**

This paper identifies and tries to address a weakness in the SWE-Bench family of benchmarks: the test suites inherited from the original pull requests are designed to verify a specific patch rather than to discriminate between correct and incorrect solutions. The authors formalize two types of test insufficiency — coverage gaps (tests fail to exercise patch-affected code regions) and semantic blind spots (tests accept superficially correct but semantically wrong patches) —  and propose SWE-ABS, a two-stage adversarial framework to strengthen test suites. Stage I uses program slicing to identify patch-relevant code, then generates coverage-guided tests with a "test decoupling" step to avoid overfitting to the gold patch. Stage II synthesizes LLM-generated semantic mutants (plausible but incorrect patches) that survive the existing tests, then produces targeted adversarial tests to kill them.

**Compliance With Llm Reviewing Policy:**

Affirmed.

**Key Questions For Authors:**

- Q1. How does SWE-ABS compare to SWE-Bench+ [1] and PatchDiff [2]? Both address the same problem — weak tests inflating SWE-Bench scores. SWE-Bench+ reports up to
  36.27 pp drops via TestEnhancer; PatchDiff finds 29.6% behavioral divergence. Can you position SWE-ABS against these works, and ideally provide empirical
  comparison on overlapping instances?

- Q2. What is the false-negative rate without test decoupling? The paper reports 4.8% after decoupling, but Table 3 does not isolate decoupling's effect.

- Q3. Does the strengthening disproportionately affect certain agent architectures? The leaderboard reshuffling is large (TRAE 1→4, live-SWE-agent 2→1). Is there a pattern — e.g., agents with more sophisticated reasoning fare better under SWE-ABS than agents that pattern-match? If so, this would support the claim that SWE-ABS provides a more faithful evaluation rather than introducing a different bias.

[1] Aleithan et al. "SWE-Bench+: Enhanced Coding Benchmark for LLMs." ICLR 2025. arXiv:2410.06992.

[2] Wang, Pradel & Liu. "Are Solved Issues in SWE-bench Really Solved Correctly?" ICSE 2026. arXiv:2503.15223.

**Limitations:**

yes

**Strengths And Weaknesses:**

## Stengths:
Strong empirical results: 24.2x improvement over UTBoost in strengthened instances, 15.75 pp average resolve-rate drop (vs. 0.70 for UTBoost), and the dramatic leaderboard reshuffling (TRAE dropping from 1st to 4th, live-SWE-agent rising from 2nd to 1st) make a compelling case. The counterintuitive finding that task difficulty ≠ test strength (SWE-Bench Pro shows comparable strengthening rates despite much lower baseline scores) is an insightful contribution.



## W1. Incomplete related work coverage.
The paper positions itself as the first systematic approach to adversarial benchmark strengthening for SWE-Bench. However, several highly relevant works are not cited or discussed.

 - SWE-Bench+ [1] addresses essentially the same problem. Its TestEnhancer component uses LLM-based analysis to identify and strengthen weak test suites on SWE-Bench, reporting resolution-rate drops of up to 36.27% on SWE-Bench Verified.
- PatchDiff [2] uses differential testing between gold patches and agent-generated patches to reveal that 29.6% of plausible patches exhibit behavioral divergence from ground truth, and that 82.7% of these divergences are invisible to existing developer tests. The problem framing ("solved issues are not really solved") and the quantitative findings are strikingly close to SWE-ABS's narrative.

The two core components of SWE-ABS — coverage-guided test generation and mutation-driven adversarial testing — both have well established precedents.
- Coverage-guided LLM test generation has been explored by CoverUp [3], which uses iterative coverage feedback loops to reach 80% median line+branch coverage, and SymPrompt [4], which leverages execution paths and AST parsing for code-aware test prompting.
- The mutation-driven adversarial loop was formalized by MuTAP [5], which achieves 93.57% mutation score at the function level.

SWE-ABS's Stage II differs primarily in the application domain (repository-level benchmark strengthening vs. function-level unit testing), but these works are worth discussing.


## W2. The UTBoost comparison may be unfair.

 The paper reports UTBoost strengthens only 10/500 (2%) instances, but UTBoost itself reports 26 instances with test insufficiency and 92 erroneous patches on
  SWE-Bench Verified. The gap likely comes from SWE-ABS evaluating UTBoost's tests only against top-30 leaderboard agents, a narrower pool than UTBoost's original
  evaluation. This should be explicitly explained in the paper.

## W3. Heavy reliance on gold patch limits applicability.
SWE-ABS requires access to the gold patch for test generation. The authors acknowledge this in Section A.12. but this significantly constrains the scope of impact.

## W4. Test decoupling effectiveness is not directly evaluated.
The test decoupling module (Section 3.1.2) is presented as a key design for preventing overfitting to the gold patch, but its contribution is not verified in the ablation. Table 3 compares Initial vs. Coverage vs. Coverage + Mutation. The missing experiment is Coverage without decoupling.

## W5. Single base model for main results.
All main results use GPT-5 as the base model. The GLM-4.7 validation (Table 10) covers only 50 instances. Since many evaluated agents are also GPT-based, GPT-5 may be better at generating mutants that exploit blind spots shared by GPT-based agents. The authors should consider adding other models as ablation.

## W6. TypeScript strengthening rate is notably low (Presentation/Soundness).
 On SWE-Bench Pro, TypeScript has an 18.18% strengthening rate (2/11) compared to Python's 75.38% (Table 7). The paper does not discuss whether this reflects a
  fundamental characteristic of TypeScript (e.g., its type system already prevents many semantic errors at compile time) or a tooling limitation in the pipeline's
  coverage and slicing support for TypeScript.

[1] Aleithan et al. "SWE-Bench+: Enhanced Coding Benchmark for LLMs." ICLR 2025. arXiv:2410.06992.

[2] Wang, Pradel & Liu. "Are Solved Issues in SWE-bench Really Solved Correctly?" ICSE 2026. arXiv:2503.15223.

[3] Pizzorno & Bader. "CoverUp: Coverage-Guided LLM-Based Test Generation." FSE 2025. arXiv:2403.16218.

[4] Ryan et al. "SymPrompt: Code-Aware Prompting for LLM-Based Test Generation." FSE 2024. arXiv:2402.00097.

[5] Dakhel et al. "MuTAP: Effective LLM-Based Test Generation via Surviving Mutant Feedback." IST 2024. arXiv:2308.16557.

---

> ### Author Rebuttal · Authors · 2026-03-29
>
> We thank Reviewer J8R3 for the thorough review. We address each concern below.
>
> **W1/Q1: Related work gaps (SWE-Bench+, PatchDiff, CoverUp/SymPrompt/MuTAP)**
>
> We will add all missing citations.
>
> *vs. SWE-Bench+/TestEnhancer.* TestEnhancer uses regex-based diff parsing to locate patch-relevant functions and generates coverage-targeted tests, but lacks program slicing, adversarial mutation, or overfitting control. We ran SWE-ABS on the same three agents evaluated by SWE-Bench+ (per-agent results in Table R1: https://anonymous.4open.science/r/B5EAafghdsg/rebuttal_tables.md). The average drop is **11.2 pp** (SWE-ABS, filtered) vs. **36.3 pp** (TestEnhancer, no filtering). One plausible explanation is that SWE-Bench+ does not report a decoupling or filtering step comparable to ours, so its reported drops may include both genuine strengthening and overfitting effects — making the two figures not directly comparable. Our W4/Q2 ablation (below) corroborates this interpretation.
>
> *vs. PatchDiff.* PatchDiff is post-hoc and per-submission: it requires an actual agent patch to diagnose divergence. SWE-ABS generates reusable tests from the gold patch alone, before any agent submission. On the same three agents evaluated by PatchDiff (per-agent results in Table R2: same link), both methods yield comparable average drops (**15.9 pp** vs. **17.3 pp**), consistent with SWE-ABS not introducing a qualitatively different ranking signal.
>
> *vs. CoverUp/SymPrompt/MuTAP.* These works operate at function level; SWE-ABS applies program slicing and adversarial mutation at repository level with gold-patch decoupling — a non-trivial adaptation required for real-world benchmark strengthening.
>
> **W2: UTBoost 10/500 vs. its own reported numbers**
>
> We applied all of UTBoost's released test suites to our top-30 agents under a unified criterion (an instance is strengthened if ≥1 agent patch is rejected): **10/500**. The top-30 are all post-2025 submissions that UTBoost never evaluated. We will clarify this in the paper.
>
> **W3: Gold patch dependency**
>
> SWE-ABS is designed for offline benchmark curation where gold patches are available by construction. Please see our response to Reviewer H9Tx (W1) for details.
>
> **W4/Q2: Missing ablation for test decoupling; false-negative rate without decoupling**
>
> As a partial check, we ran the full pipeline with and without test decoupling on a randomly sampled 100-instance subset (seed=10):
>
> | Setting | Strengthened | Avg Drop | Overfit cases |
> |---|---|---|---|
> | Full pipeline w/o decoupling | 59/100 | 24.40% | 17 |
> | Full pipeline w/ decoupling | 57/100 | 20.50% | 9 |
>
> Removing decoupling yields marginally more strengthened instances and a higher apparent avg drop, but overfit cases nearly double (9→17), indicating the additional rejections largely reflect overfitted tests rather than genuine strengthening. The false-negative rate with the full pipeline on all 500 instances is **4.8%** (24/500, Section 4.3.1). We will add this ablation to Table 3 in the revision.
>
> **W5: Single base model (GPT-5)**
>
> We extended the GLM-4.7 validation in Table 10 from 50 to 150 instances (budget-constrained). GPT-5 and GLM-4.7 produce nearly identical results: **83/150** vs. **86/150** strengthened, with drops of **19.11 pp** vs. **18.56 pp** — GLM-4.7 strengthens *more* instances than GPT-5. Results across two model families suggest SWE-ABS performance is not uniquely tied to GPT-5, partially alleviating the shared-bias concern.
>
> **W6: Low TypeScript strengthening rate**
>
> The lower TypeScript rate is better explained by sample composition: all 20 TypeScript instances come from a single repository (tutao/tutanota), making it impossible to disentangle language-level from repository-level effects. Within Python itself, scikit-learn achieves only 18.8% (6/32) on SWE-Bench Verified (Appendix Table 6), indistinguishable from the TypeScript figure. Go, also statically typed, achieves 62.5% on SWE-Bench Pro — comparable to Python (75.38%), suggesting type systems do not suppress strengthening. Manual review of 9 unstrengthened TypeScript instances: 8/9 surviving patches already satisfied augmented tests' behavioral targets; 1/9 reflected tooling limitations. We will add this discussion.
>
> **Q3: Does strengthening disproportionately affect certain agent architectures?**
>
> Yes, the pattern correlates with reasoning depth. Agents using majority voting over homogeneous parallel runs (e.g., all three TRAE submissions drop 17.00, 18.40, and 17.20 pp) structurally reinforce shared blind spots rather than correct them. In a controlled comparison, mini-SWE-agent drops **17.80 pp**, while live-SWE-agent — which builds on mini-SWE-agent by adding step-level self-reflection and autonomous tool synthesis — drops only **12.80 pp** with the same LLM (Gemini 3 Pro Preview). The 5 pp gap is consistent with SWE-ABS measuring genuine semantic correctness rather than introducing a new bias. A more systematic analysis will be included in the revision.

---

> > ### Author Rebuttal · Reviewer_J8R3 · 2026-04-03
> >
> > Thank you for the direct comparison. The overfitting explanation for the gap between 11.2pp and 36.3pp is plausible but remains a hypothesis. Could you provide concrete examples where SWE-Bench+ tests fail on semantically correct non-gold-patch solutions?  This is my primary remaining concern.

---

> > > ### Author Response · Authors · 2026-04-03
> > >
> > > Thank you for your response. We did not include concrete examples primarily because SWE-Bench+’s augmented tests and generation code are not publicly available for direct audit.
> > >
> > > In our own experiments, we observe that LLM-generated test cases can sometimes overfit to the gold patch, which can inflate the apparent drop by introducing false negatives. This observation motivates our test decoupling module, which is specifically designed to mitigate such overfitting.
> > >
> > > Given that SWE-Bench+ does not explicitly address this issue, we hypothesize that similar overfitting behavior may also exist in its test construction, potentially affecting evaluation reliability. For this reason, we do not view the two drops as directly comparable. We hope this addresses your primary remaining concern.

---

### Decision · Program_Chairs · 2026-04-30

**Decision:**

Accept (regular)

**Comment:**

This paper received mixed ratings, and at the end of the discussion two Reviewers are suggesting Weak Accept and two Weak Reject.

Reviewer MWa3's remaining concern is the technical novelty, but they are "ok with accepting if strong campaign".

Reviewer jT1K's remaining concern is that "the definitions of over-specification and error modes, which is quite core to the paper, could be more rigorously validated by human case studies," specifically arguing that "the gold experiment that would refute any suspicion of over-rejection is to perform this pipeline on multiple, independently written gold patches that reflect correctness." This constitutes the main remaining concern, since other Reviewers' concerns are not preventing them from accepting this paper.

On the positive side, among other strengths, Reviewers praised the empirical results (J8R3, jT1K) and the importance of the problem (H9Tx, MWa3). SWE-Bench is indeed widely used by the community.

While I definitely agree that carrying out a multi-gold-patch validation study will make a more robust paper, I do not think that its lack should prevent this paper from being accepted. In the borderline status of this paper, I am leaning more towards acceptance.